# Edematous severe acute malnutrition is characterized by hypomethylation of DNA

Katharina V. Schulze [1,2], Shanker Swaminathan[1,2], Sharon Howell[3], Aarti Jajoo[1,2], Natasha C. Lie[2,4], Orgen Brown[3], Roa Sadat[2], Nancy Hall[2], Liang Zhao[5], Kwesi Marshall[3], Thaddaeus May[2], Marvin E. Reid[3], Carolyn Taylor-Bryan[3], Xueqing Wang[1,2], John W. Belmont [1,2], Yongtao Guan[1,2], Mark J. Manary[6,7], Indi Trehan [6,7,8], Colin A. McKenzie[3,9] & Neil A. Hanchard [1,2]*

Edematous severe acute childhood malnutrition (edematous SAM or ESAM), which includes kwashiorkor, presents with more overt multi-organ dysfunction than non-edematous SAM (NESAM). Reduced concentrations and methyl-flux of methionine in 1-carbon metabolism have been reported in acute, but not recovered, ESAM, suggesting downstream DNA methylation changes could be relevant to differences in SAM pathogenesis. Here, we assess genome-wide DNA methylation in buccal cells of 309 SAM children using the 450 K microarray. Relative to NESAM, ESAM is characterized by multiple significantly hypomethylated loci, which is not observed among SAM-recovered adults. Gene expression and methylation show both positive and negative correlation, suggesting a complex transcriptional response to SAM. Hypomethylated loci link to disorders of nutrition and metabolism, including fatty liver and diabetes, and appear to be influenced by genetic variation. Our epigenetic findings provide a potential molecular link to reported aberrant 1-carbon metabolism in ESAM and support consideration of methyl-group supplementation in ESAM.

[1] Department of Molecular and Human Genetics, Baylor College of Medicine, Houston, TX, USA. [2] USDA/ARS/Children's Nutrition Research Center, Baylor College of Medicine, Houston, TX, USA. [3] Tropical Metabolism Research Unit, Caribbean Institute for Health Research, University of the West Indies, Mona, Jamaica. [4] Graduate Program in Integrative Molecular and Biomedical Sciences, Baylor College of Medicine, Houston, TX, USA. [5] Precision Medicine Research Center, Taihe Hospital, Shiyan City, China. [6] Departments of Paediatrics and Child Health and Community Health, University of Malawi, Blantyre, Malawi. [7] Department of Pediatrics, Washington University in St. Louis, St. Louis, MO, USA. [8] Departments of Pediatrics and Global Health, University of Washington, Seattle, WA, USA. [9]Deceased: Colin A. McKenzie *email: hanchard@bcm.edu

Annually, severe acute malnutrition (SAM) directly contributes to nearly one million deaths of children under age 5 globally, and indirectly to millions more[1]. Despite concordant micro- and macro-nutrient deficiencies, SAM classically presents as one of two phenotypically distinct forms—edematous SAM (ESAM), which includes the syndromes of kwashiorkor and marasmic-kwashiorkor, and nonedematous SAM (NESAM) or marasmus[2,3]. NESAM is typically characterized by weight loss and wasting, while ESAM is defined by the presence of bilateral, pitting edema and usually has more overt, and often more severe, multiorgan dysfunction, including hepatic, hematopoietic, and gastrointestinal impairment, alongside skin and hair abnormalities[4–7].

Given that ESAM accounts for 50–70% of SAM cases in some developing countries[8], and has different clinical outcomes relative to NESAM, considerable interest remains in gaining a better understanding of the underlying molecular pathophysiology[2,9]. The divergent SAM phenotypes are not fully explained by environmental, dietary, or infectious factors[7,10–12], despite a substantial catalog of disparate studies describing biochemical and physiological differences between the two forms[13–16], including recent reports of quantitative shifts in microbiota[17,18]. These observations, however, are subject to reverse causation and provide an incomplete picture of the broad pathophysiological dysfunction that is the hallmark of ESAM[19]. Moreover, despite differences in pathophysiology and clinical outcome, the treatment of SAM has remained largely unchanged and is the same regardless of form[20–22]. Although public health strategies that address food security remain important to SAM[23], such strategies are notoriously difficult to implement and are susceptible to volatility in climate and society. A better understanding of the underlying molecular pathophysiology of ESAM could provide additional therapeutic insights that could be used to further improve outcomes and prevent development of the condition.

There are no definitive molecular or genetic correlates of acute childhood ESAM; however, ESAM and NESAM differ biochemically in the flux of constituent metabolites of the 1-carbon cycle[24,25]. During steady-state re-feeding, but while still acutely ill, ESAM patients show a significantly lower concentration of the essential amino acid methionine with slower methyl-group and total methionine flux[25] in the conversion of methionine to S-adenosyl-methionine (SAM-e). SAM-e is the major source of methyl groups required for the methylation of cellular components, including the methylation of DNA[26–28] during mitosis. These differences in 1-carbon metabolism between ESAM and NESAM, however, are not observed after recovery from SAM, implying that nutritional and dietary recovery involves a reversal of the biochemical changes observed during the acute illness. While previous studies have shown altered DNA methylation in individuals with a history of severe acute malnutrition more generally[29,30], none compared DNA methylation between the edematous and nonedematous forms of SAM, particularly during acute disease. Since the phenotypic differences that characterize the two forms of SAM are only evident during acute malnutrition, concomitant molecular changes incurred during acute illness could potentially highlight important drivers of the differing pathophysiologies. We hypothesize that in cells with high mitotic rates, acutely ill children with ESAM will have lower DNA methylation compared to their NESAM counterparts, whereas methylation differences among recovered SAM individuals will be less disparate. Given the interrelationship between DNA methylation, gene regulation, and sequence variation, we further postulate that the integration of these modalities across acute and recovered SAM participants might provide valuable insights to the distinct pathophysiology of ESAM.

Here, we systematically assess DNA methylation at 420,500 genome-wide CpG sites in buccal epithelium DNA samples obtained from two independent SAM cohorts—309 prospectively recruited, acutely ill children (Fig. 1; Supplementary Table 1) with SAM from Malawi and Jamaica and 65 adult SAM survivors from Jamaica; the latter were retrospectively recruited 16 or more years after their acute malnutrition event (Fig. 1; Supplementary Table 1). Methylation is compared between ESAM (cases) and NESAM (controls), and differentially methylated sites are then interrogated for their corresponding gene expression profiles and surrounding single nucleotide variation.

## Results

**Hypomethylation of DNA characterizes acute ESAM.** A comparison of genome-wide methylation between acutely ill children (predischarge, DC samples) with ESAM and NESAM revealed significant methylation differences at 877 CpG sites at a false discovery rate (FDR) < 0.01, of which 157 were significant at a conservative Bonferroni-corrected threshold of $P < 1.2 \times 10^{-7}$ (t test; Fig. 2a). The mean absolute difference in methylation beta-values across all significant sites was 0.054 (SD 0.02). Consistent with previous observations of slower methyl-flux in acute ESAM,

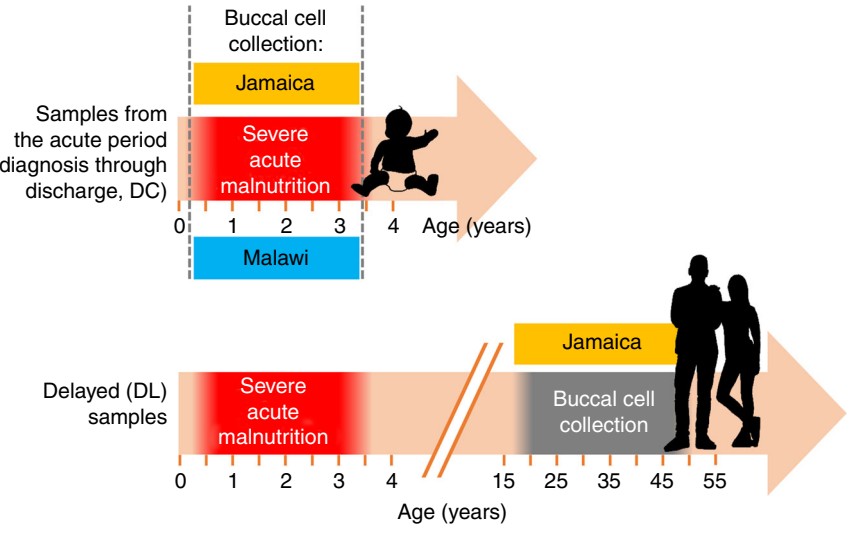

**Fig. 1** Overview of sampling timeline across both cohort locations and time points.

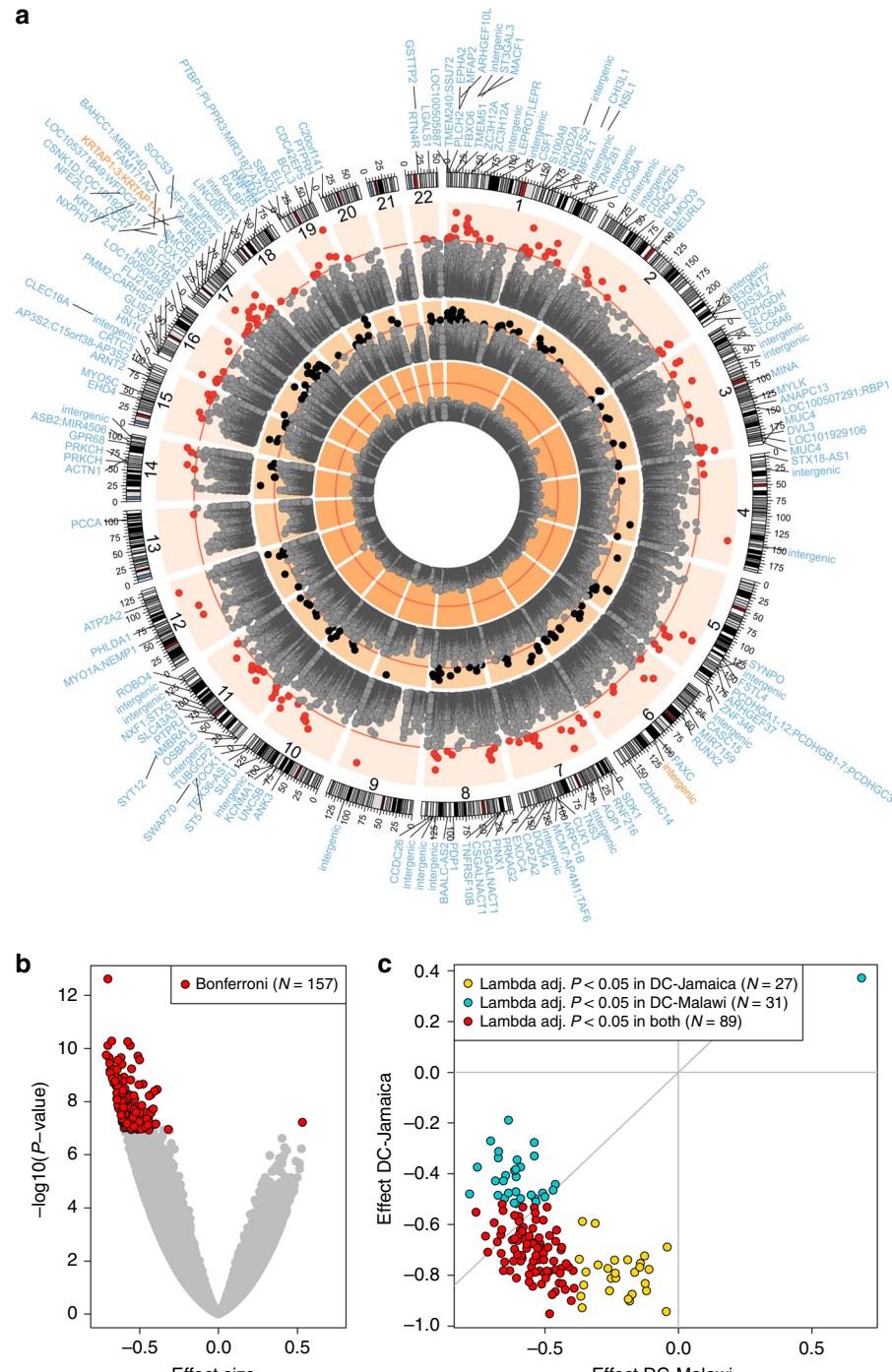

**Fig. 2 DNA hypomethylation in ESAM. a** The genome position of each locus is plotted against its $-\log_{10}(P$ value). Depicted, from the inner to the outermost ring, are the recovered (DL) single site, acute (DC) single site, and DC cluster-based differential methylation results. Red lines mark the Bonferroni threshold of significance for each analysis. Black ($N = 157$) and red dots ($N = 166$) pass this significance threshold, in their respective analyses. Gene symbols around the plot perimeter represent the gene(s) within 10 kb of the significantly differentially methylated clusters (outer ring). Gene symbols in blue are hypomethylated in ESAM, gene symbols in orange are hypermethylated. **b** Volcano plot of effect sizes among all DC single sites. **c** Concordance of effect size and adjusted statistical significance between Jamaica and Malawi DC samples at the $N = 157$ Bonferroni-significant single CpG sites; ten sites were found to be age-sensitive in each of the country-specific analyses and were omitted (Methods). All $P$ values are based on $t$ test results from regression analyses.

all but one differentially methylated site (156/157; 99%) were significantly hypomethylated in ESAM relative to NESAM (Fig. 2b). Differentially methylated sites showed the same direction of effect (hypo- or hypermethylation) with a consistent effect size (binomial test, $P = 3.08 \times 10^{-14}$) and magnitude ($t$ test,

lambda adjusted $P < 0.05$ in both cohorts for 89 of 147 probes (60%)) in both Jamaican and Malawian samples (Fig. 2c; Supplementary Data 1). Conversely, among samples taken from adult Jamaican survivors of SAM several years after the acute insult (delayed, DL samples), we found no evidence for differential

methylation between ESAM and NESAM using either our strin-
gent Bonferroni-corrected cut-off or the more permissive FDR <
0.01 for significance (Fig. 2a), despite being adequately powered
to detect the magnitude differences observed between DC sam-
ples. This was again consistent with the previously observed lack
of differences in methyl-flux between ESAM and NESAM at the
time of recovery.

Methylation levels at adjacent CpG sites show strong
correlation[31]; therefore, to bolster support for differential
methylation of significant single CpGs and to limit potential
false-positives, we divided the genome into windows of clustered
CpGs in which adjacent CpGs were within 10 kb and showed the
same direction of effect (Methods). We then compared methyla-
tion in ESAM and NESAM averaged across the resulting regions
using the same parameters and covariates as our single-site
analysis. Reassuringly, 102 of the 157 single sites (65.0%) were
found in the resulting 166 Bonferroni-significant differentially
methylated clusters (DMCs; $t$ test, $P < 3.7 \times 10^{-7}$, corrected for
135,053 clusters). The majority of the remaining significant single
sites (50/55; 91%) fell into clusters that were significant at FDR <
0.01 (Supplementary Data 2). An additional 71 Bonferroni-
significant DMCs had no individual CpGs meeting our single-site
cut-off for significance, but, as a cluster, showed significant
differential methylation between the two groups. All but two of
the 166 significant DMCs were hypomethylated in ESAM
compared to NESAM (Fig. 2a).

**Hypomethylation has varying effects on gene expression**. In
order to better understand the potential regulatory effect of the
differentially methylated loci, we next looked for genic features
that were overrepresented among the CpG sites found within
significant DMCs, relative to all tested loci. CpGs within DMCs
were significantly enriched over gene bodies (i.e. coding exons
and intervening introns; hypergeometric test, $P < 1.4 \times 10^{-6}$) and
relatively depleted in regulatory regions upstream of the tran-
scription start site (TSS) and over the first exon (Fig. 3a).
Moreover, the effect sizes obtained from the single site linear
regression analysis appeared to differ modestly by gene context
annotation (Kruskal–Wallis test, $P = 0.047$; Supplementary
Fig. 1), with smaller effect sizes at loci within 200 bp of tran-
scription start sites relative to those of CpG loci within 1500 bp of
the TSS (Dunn's test, $P = 0.004$), gene bodies (Dunn's test, $P =
0.012$), 5′ untranslated regions (UTR; Dunn's test, $P = 0.019$), and
the first exon (Dunn's test, $P = 0.022$). Conversely, the magni-
tudes of effect sizes of Bonferroni-significant loci found within
1500 bp of TSSs were larger than those of CpG loci in intergenic
regions (Dunn's test, $P = 0.013$).

The regulatory consequences of methylation on gene activity
vary[32], such that the effect of hypomethylation on gene
expression is dependent upon genomic and tissue context;
therefore, to provide further context for our findings, we assessed
intraindividual correlation between gene expression and genome-
wide DNA methylation levels in buccal cell samples obtained at
the time of diagnosis from 20 additional Malawian children.
Correlation was determined between CpG methylation and
expression of highly expressed genes (expression levels twofold
above the mean antigenomic background; Methods) found within
10 kb of our DMCs ($N = 25$; Supplementary Data 2; Fig. 3b).
With the exception of five CpG probes binned within one DMC
that overlapped two highly expressed genes, *PMM2* and
*CARHSP1*, all methylation values were correlated with the
expression of a single gene. Consistent with the notion that
promoter methylation is generally repressive of gene expression,
correlations at methylation sites within 200 bp of the transcrip-
tion start site (TSS200) tended to be inversely related to gene

expression (two-sided Wilcoxon signed rank test, $P = 0.027$);
negative correlation between methylation and *MED24* gene
expression is exemplified in Fig. 3c. Conversely, methylation at
intergenic regions (IGR) was generally positively associated with
gene expression (two-sided Wilcoxon signed rank test, $P =
0.030$). However, at other gene regions, and particularly at gene
bodies, CpG methylation showed significant positive and negative
correlations with gene expression (Fig. 3b; Supplementary Data 3),
suggesting a more complex transcriptional response to the acute
insult than might have been anticipated from widespread
hypomethylation.

**Differentially methylated loci relate to nutrition and
metabolism**. Next, we sought to understand the potential disease
relevance of genes within our DMCs. Given the lack of known
genetic loci or molecular correlates for ESAM, we leveraged
publicly available databases and ontologies to provide disease and
molecular context to our hypomethylated genes.

We first interrogated the catalog of genome-wide association
studies (GWAS)[33] for genes within 10 kb of our significant DMCs
that were also implicated by replicated GWAS SNPs of genome-
wide significance. We then agnostically projected the resulting
gene list on to disease traits using their mapped experimental
factor ontologies (EFOs, Methods). Thirty-two EFOs were
represented at least twice by 33 of the 237 genes found within
10 kb of our Bonferroni-significant hypomethylated loci (Fig. 4a).
Some of these EFOs occurred frequently when we repeatedly
resampled 237 of the 2436 genes in the GWAS catalog that are
also targeted by the 450 K array (April 24, 2018); for example,
body height was enriched by the random gene set in 93% of
10,000 permutations (Supplementary Fig. 2). This is likely
representative of the number of GWAS that address body height
as well as the number of genes associated with the trait. However,
mapping 33 GWAS catalog genes to 32 associated EFO terms, as
seen in our data, was observed with a probability of only 0.06
among random permutations (i.e. ≥33 GWAS genes mapping to
≥32 EFO terms). Nutrition and cardiometabolic disease were
among the enriched parent ontologies shared across EFO terms
linked to our Bonferroni-significant genes (color-shaded in
Fig. 4a).

To further explore the disease relevance of our hypomethylated
loci, we also attempted to place our top candidate genes in the
context of the organ-specific phenotypes of ESAM using the
human phenotype ontology (HPO) (Methods)[34,35], which
includes 34 phenotypes associated with the term kwashiorkor
(Fig. 4b, Supplementary Data 4). These kwashiorkor-related
HPOs, in turn, were associated with 1747 genes. Twenty
kwashiorkor HPO genes were also among the 237 Bonferroni-
significant DMC genes, while 143 overlapped the 1549 genes
associated with FDR < 0.01 significant DMCs. We randomly
selected 10,000 different gene combinations from the set of 24,094
genes within 10 kb of all regions tested for differential methyla-
tion that were equal in size to either the number of Bonferroni or
FDR-significant genes; in each instance we then determined the
number of randomly selected genes overlapping genes linked to
kwashiorkor HPOs. Based on the resulting distributions, we
found that the number of genes overlapping between true DMC
genes and kwashiorkor HPO genes was larger than random by
approximately one standard deviation for Bonferroni-significant
genes ($Z$-score = 1.11; Supplementary Fig. 3a) and more than
four standard deviations for FDR-significant genes ($Z$-score =
4.19; Supplementary Fig. 3b).

To date, however, only a minority of the >20,000 human genes
have been mapped to HPO terms ($N = 3498$, build #129), largely
on the basis of phenotypes observed in single-gene Mendelian

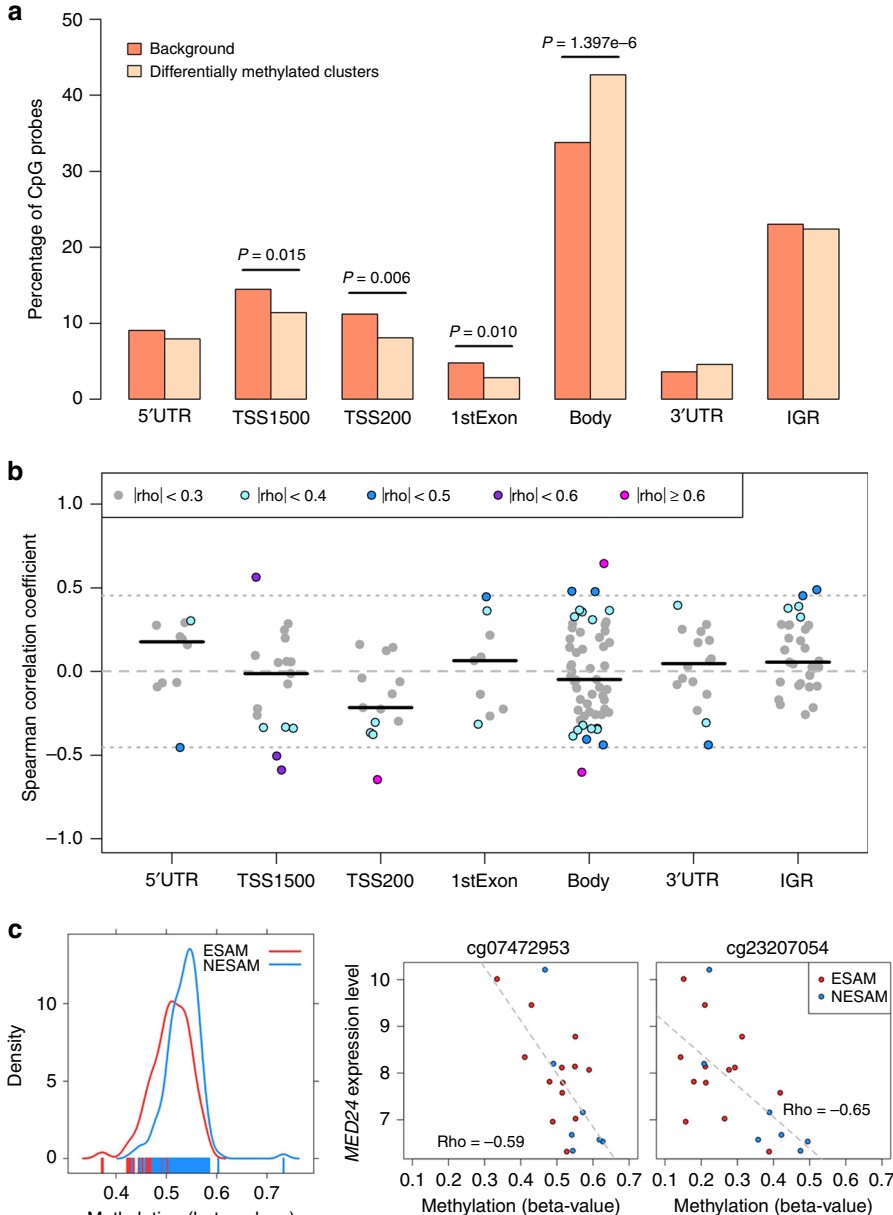

**Fig. 3 Gene feature enrichment and expression analysis. a** Gene annotation enrichment analysis for CpG probes within significant differentially methylated clusters (DMCs) from the analysis of acutely malnourished children. $P$ values were calculated with hypergeometric tests of depletion or enrichment comparing $N = 420{,}500$ of all tested, background CpG probes with the subset of $N = 630$ CpG probes found within DMCs. **b** Intraindividual correlation (Spearman) between mean methylation and mean expression of highly expressed genes within 10 kb in 20 additional Malawian children with SAM at individual CpG probes within significant DMCs. Horizontal dark bars represent median correlation coefficients, dashed line shows correlation of zero, and dotted lines indicate thresholds beyond which correlations reach statistical significance (algorithm AS 89 $t$ test, $P < 0.05$). 5′UTR: 5-prime untranslated region, TSS1500: within 1500 bp distance to transcription start site, TSS200: within 200 bp distance to transcription start site, 3′UTR: 3-prime untranslated region, IGR: intergenic region. **c** Beta-value methylation levels by SAM type in DMC overlapping *MED24* (left panel, $N_{ESAM} = 164$ samples, $N_{NESAM} = 145$ samples) next to scatterplots showing the relationship between methylation levels of probes within the *MED24* containing DMC and the expression values of *MED24* (middle and right panel). Gray dashed lines show a linear model fit, while rho values indicate Spearman correlation coefficients. Source data for panels **a** and **c** (density plot) are provided as Source Data file.

disorders. Therefore, to better understand the link between our candidate genes and kwashiorkor HPOs, we used gene ontologies, which collate genes with similar properties, as intermediate links between HPO-annotated genes and our ESAM genes (Supplementary Fig. 4).

The 1748 genes annotated to HPO terms associated with kwashiorkor were significantly enriched (hypergeometric test, FDR < 0.01) in 2464 gene ontologies—we refer to these as

kwashiorkor-phenotype gene ontologies (KGOs). We then assessed how many of the 64 gene ontologies enriched (hypergeometric test, $P < 0.01$ vs. background) by the 237 genes in our Bonferroni-significant DMCs (Supplementary Data 5) were also in the list of KGOs. Just over half (33/64; 50.7%) of these study gene ontologies (SGOs) were also KGOs, and the majority of these (29/33; 88%) were also in the top quartile for significance in the KGO list, implying that the gene ontologies

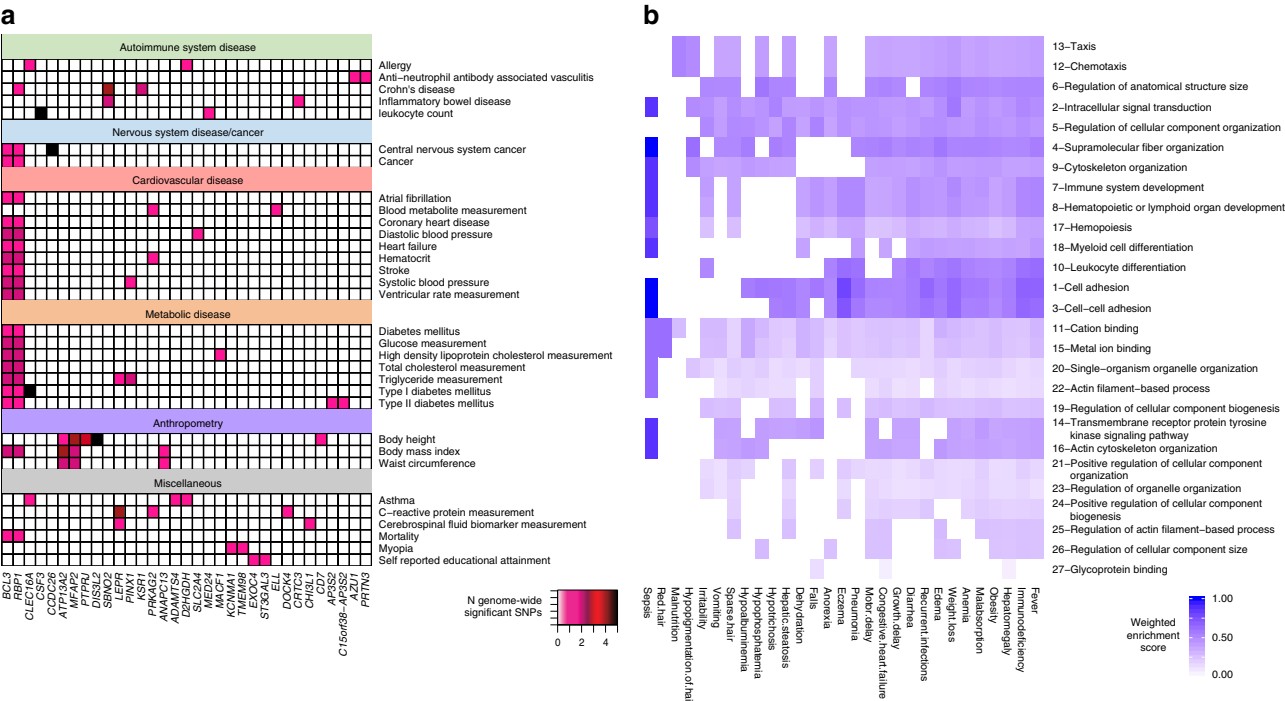

**Fig. 4 Disease gene enrichment in ESAM. a** Experimental Factor Ontologies (EFOs) from the GWAS catalog (Methods) with two or more genes represented among ESAM differentially methylated clusters (DMCs) are shown on the vertical axis and grouped according to their shared parent EFOs. The overlap between disease and gene is indicated by a shaded box. The color key associated with each box represents the number of genome-wide significant SNPs for each gene for the listed EFO. **b** Weighted enrichment values for gene ontologies enriched by study DMCs (SGOs; vertical axis) and phenotypes associated with kwashiorkor (KGOs; horizontal axis) in the Human Phenotype ontology (HPO). SGOs that are not found in the KGO list receive a value of 0 by default, and SGOs that are found in the KGO list and in the top quartile of both lists would have a value of 1. Ranking of SGOs is indicated in descending order by the number in front of the SGO term and was based on their aggregate weighted enrichment score.

most highly enriched by our study genes were also highly enriched among the kwashiorkor phenotypes. We quantified the overlap between the two gene ontologies by creating a weighted enrichment score for SGOs that was maximized by SGOs that both overlapped the KGO list and had a high significance rank among KGOs (Methods). The kwashiorkor subphenotypes of immunodeficiency and fever, followed by hepatomegaly and obesity had the strongest SGO scores, again reflecting the significance of metabolism, immunity, and the liver to ESAM pathophysiology (Fig. 4b).

Closer scrutiny of the individual genes encompassed by our DMCs revealed several candidates with potential importance to nutrition in general and the ESAM phenotype in particular. For instance, hepatic steatosis (fatty liver) is associated with obesity but is also well-described in ESAM[36–38], with severe involvement at the time of diagnosis heralding a poor prognosis[39]. Our list of Bonferroni-significant hypomethylated genes included *NDUFS2* (NADH:Ubiquinone oxidoreductase core subunit S2; Supplementary Fig. 5a), which encodes a subunit of mitochondrial complex I that has recently been implicated in the development of malnutrition-related fatty-liver in rats[37]. *PHLDA1* (Pleckstrin Homology Like Domain Family A Member 1), which encodes a nuclear protein associated with adipogenesis, was also in one of our top DMCs; hypermethylation of hepatic *Phlda1*, with concomitant transcriptional silencing, has been reported in mice who developed fatty liver after being fed a high-fat diet[40]. Other DMCs included *NFE2L1*, a nuclear transcription factor integral to the maintenance of proteasome function in hepatocytes that has also been implicated in liver steatosis[41], and *SLC2A4*, which encodes an insulin-regulated facilitative transporter of glucose (GLUT4) that leads to hepatic steatosis in homozygous knockout mice[42].

One of the largest and most significantly hypomethylated DMCs included *SOCS3* (Supplementary Fig. 5b), a regulator of cytokine-, JAK-STAT-, and leptin- signaling, for which changes in methylation have been associated with metabolic syndrome[43], body mass index (BMI)[44], and diabetes[45], while changes in expression in muscle tissue have been associated with diet-induced metabolic derangements[46]. These observations, alongside the occurrence of several other inflammatory and immunoregulatory genes in our top DMCs, including *LENG8, CHI3L1, SDK1*, and *CSF3* (Supplementary Fig. 5c), provide a potential pathophysiological link with the altered inflammatory state noted in ESAM[47]. Other DMCs included the *O*-linked glycosylation pathway enzyme genes *B3GNT7* (Supplementary Fig. 5d), *CSGALNACT1*, and *ST3GAL3*, which may be relevant to observations of reduced sulfated glycosaminoglycans in intestinal biopsies from children with acute kwashiorkor[16], and *AQP1* (Aquaporin 1), a member of the aquaporin family of transcellular water channels[48] that has been implicated in the development of tissue edema. A potential role for aquaporins was further supported by an FDR-significant DMC ($t$ test, $P = 4.52 \times 10^{-5}$, FDR $= 6.42 \times 10^{-3}$) that included two other members of the same family (*AQP5* and *AQP6*; Supplementary Data 2).

**Potential role for sequence variation in acute ESAM.** Given the established association between single nucleotide polymorphisms (SNPs) and methylation at nearby CpG sites (methylation quantitative trait loci—meQTLs)[49–51], we also sought to determine whether genetic variation in and around our hypomethylated loci might influence the extent of methylation at our DMCs. A model of genetic influence in SAM, however, would need to account for our observation of differential methylation between

ESAM and NESAM only being evident during the acute nutritional stress. Context-dependent expression quantitative trait loci (eQTLs) have been described in the literature[52–54], and, although similar observations for meQTLs are few, both eQTLs and meQTLs are thought to represent binding sites for transcription factors (TFs), with meQTLs influencing the activity of core methylation enzymes[55] instead of directly impacting gene transcription (eQTLs). In the context of acute SAM, therefore, in which we had some evidence for a complex transcriptional state, we considered whether there was any evidence for nutritional-state-dependent meQTLs. We explored this by integrating methylation data with genome-wide genotyping data available for 138 SAM samples from Jamaica (Methods) and evaluating meQTLs at our DMCs in the context of acute (DC) and recovered (DL) samples. Several significant (Wald test, $P < 3.4 \times 10^{-8}$) meQTL signals were observed in both DC (5030 SNP−CpG pairs) and DL (1346 SNP−CpG pairs) sample groups (Supplementary Fig. 6).

We focused on meQTLs within 10 kb of any of the 1261 DMCs surpassing our FDR threshold ($t$ test, FDR < 0.01), and, to account for subdividing of our initial sample, we first considered all SNP−CpG associations of nominal significance (Wald test,

$P \le 0.05$; Methods). At these FDR DMCs, 1885 of the 16,792 SNP-CpG pairs met our significance threshold among acute (DC) samples; of these, 162 pairs (~8.5%), representing 111 DMCs (Supplementary Data 6), had evidence of a significant difference between ESAM and NESAM in the strength of meQTL association (Wald test, $P_{interaction} \le 0.05$) with no evidence of a similar interaction among recovered (DL) samples (Methods). This group included 16 SNP−CpG pairs at 14 of our Bonferroni-significant DMCs (Fig. 5a). At these loci, the genotype-methylation association was significant in either acute NESAM or ESAM but was no longer significant at the time of recovery. Among the 162 disease-specific meQTLs, 26 SNP-CpG pairs (16 unique loci) were still evident at a more stringent significance threshold (Wald test, $P < 1 \times 10^{-4}$), and the majority of this latter group (23 of 26) showed strong association in acute ESAM, but were not significant in acute NESAM or among recovered groups (Fig. 5b, c). Interestingly, almost two-thirds (64.1%) of these candidate nutrition-sensitive meQTLs were predicted to strongly alter the likelihood of TF binding[56,57], although, a similar proportion was observed among putative nutrition-insensitive meQTLs (65.1%; Supplementary Data 7 and 8) and there was substantial overlap between the two sets of TFs. The ten (8.5%)

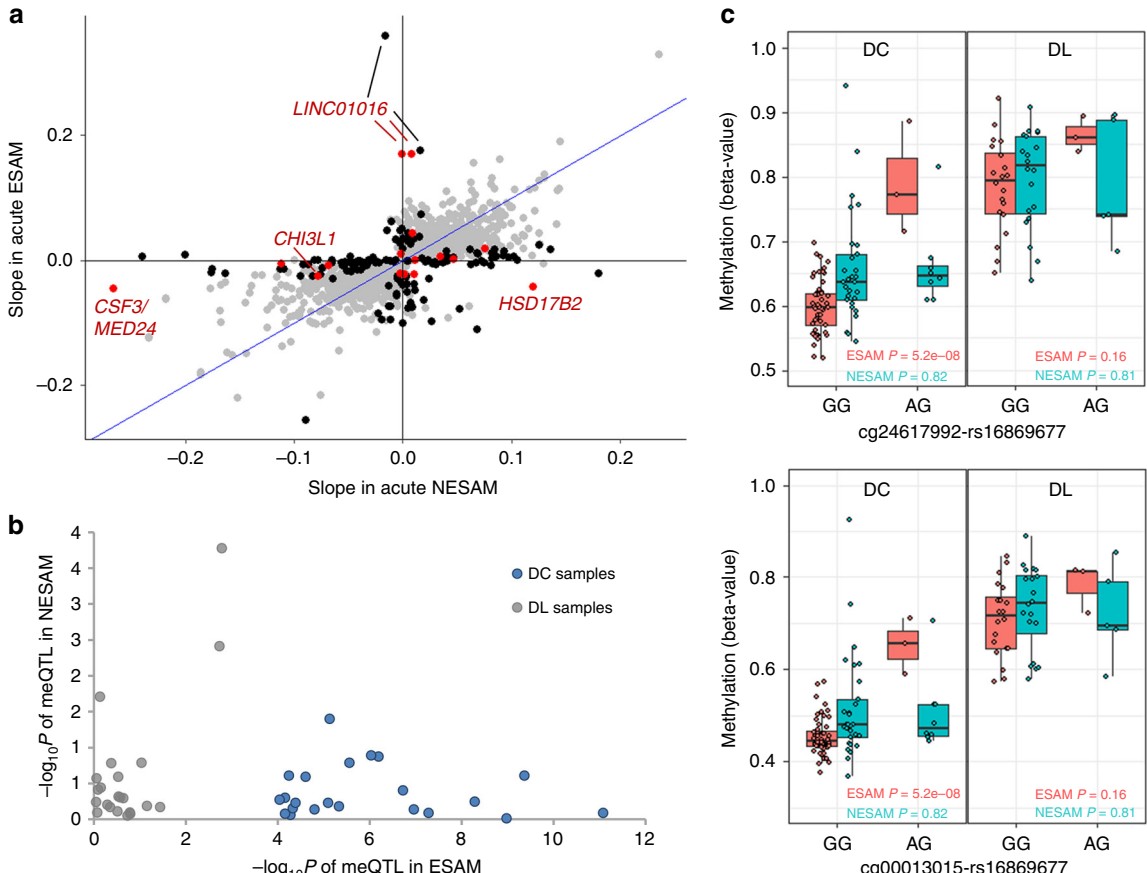

**Fig. 5 A model of nutrition-sensitive methylation quantitative trait loci (meQTLs). a** Scatterplot of the slopes of the regressions ($\beta_1$) of SNP genotype on CpG methylation (meQTLs) at significant DMCs in acute (DC) ESAM and NESAM samples ($N = 1885$ SNP−DMC pairs). FDR-significant DMCs with meQTLs showing a significant interaction ($t$ test, $P \le 0.05$) between ESAM and NESAM DC samples, but not among DL samples ($N = 162$) are highlighted in black; similar meQTLs found at Bonferroni-significant DMCs ($N = 16$) are in red. **b** $\log_{10}$ transformed $P$ values of 162 meQTLs with a significant interaction; each dot represents values for the same SNP−CpG pair between acute (DC NESAM vs. DC ESAM) and recovered (DL NESAM vs. DL ESAM) individuals. **c** Disease context-dependent meQTLs at *LINC01016*. Methylation beta values ($y$ axis) and genotypes ($x$ axis) are shown for two adjacent SNP−CpG pairs (one SNP—two CpGs) illustrating a significant ESAM−NESAM interaction ($t$ test, $P \le 0.05$; positive regression slope) in acute (DC, top and bottom left panels, $N_{ESAM} = 53$ samples, $N_{NESAM} = 37$ samples) but not recovered SAM (DL, top and bottom right panels, $N_{ESAM} = 23$ samples, $N_{NESAM} = 25$ samples). Boxplot center lines are medians, box boundaries are first and third quartile, and whiskers extend to data points found within 1.5 times the length of inter quartile range from the median.

TFs unique to the nutrition-sensitive meQTLs included the related TFs MLX and MLXIPL, which participate in pathways associated with nonalcoholic fatty liver disease (NAFLD)[58]. Given the modest samples sizes employed and a limited view of African genetic variation, however, it is likely that some of the nutrition-insensitive meQTLs could be misclassified and are really nutrition-sensitive, or that more complex mechanisms of activation in the context of severe malnutrition could differentiate the two datasets.

To further explore this nutrition-sensitive meQTL model, we evaluated our set of meQTLs against meQTLs identified in the same tissue in another childhood methylation study, which analyzed individual methylation differences in monozygotic twin pairs[59]. Only 10 (6.2%) out of 162 putative nutrition-sensitive buccal meQTLs were also found in this reference buccal meQTL dataset, in contrast to 238 (12.6%) out of 1723 non-nutrition-sensitive meQTLs (Fisher's exact test, $P = 0.005$). We also performed a similar analysis using meQTLs assessed across tissues and age-groups in the ARIES study[60]. Similar to the reference buccal meQTLs, we found that only 9 of the 162 putative nutrition-sensitive pairs (5.6%) were identified as meQTLs in any ARIES data subset, whereas almost twice the proportion (11.6%; 200 of 1723) of putative non-nutrition-sensitive meQTLs were represented (Fisher's exact test, $P = 0.018$; Supplementary Table 2); this was consistent with a model in which proposed nutrition-sensitive meQTLs remain relatively quiescent during states of adequate nutrition—that is, the effect of the SNP on methylation is minimally (or not) evident under normal circumstances—but becomes manifest during the acute nutritional stress of SAM.

## Discussion

Using genomic and epigenomic platforms that have been applied to other complex traits, we provide a molecular evaluation of ESAM. We deliberately focused on differences incurred in the context of acute starvation, such that the molecular changes described are specific to the acute pathology of ESAM, and relative to NESAM. This differentiates our study from previous studies of more general malnutrition that are agnostic to the form of SAM[29,30]. Similarly, by integrating molecular correlates of expression and assessing the contribution of cis-acting genetic variation, we provide a unique view of this gene-environment response that may hold lessons for other disorders.

Consistent with lower concentrations and slower methyl-flux of methionine reported in acute, but not recovered, ESAM, we observed extensive DNA hypomethylation in ESAM children relative to their NESAM counterparts during the acute insult that was not evident among recovered individuals. Patterns of methylation can vary widely between tissues, making extrapolation of single-tissue findings challenging. We chose buccal epithelium as an easily accessible tissue that is embryologically related to the gut, one of the major organs affected in ESAM; thus some of our DMCs may be directly relevant to the emerging literature on gut pathophysiology in ESAM[61]. Furthermore, methylation levels in buccal epithelium often correlate with methylation in other cells[62], which aids in interpreting observed DMCs as also relevant to other ESAM-disturbed organs, such as the liver. In fact, 40% of our differentially methylated sites showed strong correlation (Pearson $r \geq 0.7$) between buccal and blood DNA methylation (Methods, Supplementary Data 9). This assertion is further bolstered by the aforementioned stable-isotope studies of methionine metabolism[25], which described the slowing of whole-body methyl-group turnover in ESAM; this would be expected to impact DNA methylation patterns across the breadth of mitotically active tissues.

Hypomethylation was enriched over gene bodies and, in additional age-, gender- and SAM-status matched samples, methylation at differentially methylated loci within gene bodies correlated with both increased as well as decreased expression at proximal genes; this suggests that DNA methylation changes in acute SAM have a complex effect on neighboring gene transcription. Such a complex transcriptional profile in the context of hypomethylation is consistent with recent studies of the methylation-expression interface, which have emphasized genome context-dependent, bidirectional associations between altered DNA methylation and gene transcription[63].

Hypomethylated DMCs overlapped genes associated with organ dysfunction similar to those recorded in ESAM over decades of metabolic and clinical studies[4–6,36,64–67]. The implicated genes included disruption of pathways involved in liver pathology, inflammation, and metabolism. This latter group also included genes associated with common nutritional and cardio-metabolic disorders. Fatty liver is known to occur in both under- and overnutrition; our results suggest that this might reflect regulatory disruptions in a core set of genes, including NDUFS2 and PHLDA1, that lead to steatosis regardless of the directional effect of the nutritional insult. This was a recurrent theme with genes related to metabolic stress such as SOCS3, CHI3L1, and CSF3. Our data suggest that the same genes that are in involved in obesity/diabetes might also be relevant to ESAM, possibly causing a mirrored molecular response to the availability of nutrient energy sources that utilizes similar pathways.

Our observations may also be relevant to the chronic diseases and long-term health effects described among survivors of acute starvation[68–70]. Genes implicated in our ESAM study showed little-to-no overlap with those noted in a previous study of methylation in adult survivors of malnutrition[29]; however, that study did not include the severity or type of SAM, and primarily focused on whole blood samples, making direct comparisons difficult. Nonetheless, it may be that the underlying mechanisms postulated here are still relevant—mitotic turnover among growing children is typically higher than in adults; and, whereas buccal epithelium is dynamically replenished, other cell types demonstrate less continuous renewal, allowing aberrations in methylation incurred during childhood to be perpetuated in tissues such as heart or muscle, potentially leading to the long-term health consequences that have been associated with the divergent SAM phenotypes[30,69,70].

We also observed preliminary evidence for a model of nutrition-sensitive meQTLs at 8% of our differentially methylated loci. The diagnosis of ESAM depends upon the clinical recognition of nutritional edema; however, the occurrence (and degree) of other clinical manifestations, such as skin, hair, or liver changes, is more variable. Given the relevance of genetic variation to phenotypic variation, our speculation is that nutrition-sensitive meQTLs might fine-tune the degree of hypomethylation at DMCs, akin to genetic modifiers of disease, and thereby modulate the clinical expression of ESAM phenotypes. Given limitations in genome-wide coverage of African-ancestry populations inherent in current genotyping platforms[71], and differences in ancestral make-up between SAM cases and tissue- and age-comparable controls, larger studies utilizing longitudinal sampling in diverse disease contexts, populations, and cellular/organism models are required to determine the extent to which this hypothetical model holds true.

Our findings do not give a definitive answer to the long-standing question of the primary cause of kwashiorkor (i.e. what causes some children to develop kwashiorkor and others marasmus); however, the epigenetic changes observed provide a plausible link between observations of altered 1-carbon metabolism in acute SAM and the clinical manifestations of

ESAM/kwashiorkor. Whilst we cannot fully assay the effect of perturbed methylation at the organismal level, children with SAM are typically ill for days to weeks beforehand, suggesting the potential for a widespread, systemic shift in biology as part of the pathophysiology. Outside of cancer biology, there are few disease states in which epigenetic changes have been clearly linked to nutritional metabolism. Regulation of DNA methylation is complex and we cannot fully discount that the methylation changes observed here are purely secondary or unrelated to either 1-carbon metabolism or ESAM; however, given the interdependencies of 1-carbon metabolism, methyl-group flux and DNA methylation[72,73], our working hypothesis is that the observed DNA hypomethylation is closely related to the slow turnover of 1-carbon cycle metabolites, including methionine, noted in previous studies. The root of this turnover difference remains obscure but could reflect subtle upstream differences in diet quality or innate handling of dietary nutrients (e.g. genetic differences in metabolism) or both.

Irrespective of the initiating insult, downstream differences in pathophysiology that distinguish the two acute clinical states could still relate to the epigenetic changes observed in ESAM. In fact, a theoretical link between the biological processes regulated by 1-carbon metabolism and the organ dysfunction of ESAM was postulated decades ago[74], but never empirically tested. This provides a context for further integrating biochemical and metagenomic observations of ESAM with our findings. The levels of sulfur-containing 1-carbon amino acids cysteine and methionine in plasma have consistently been shown to be lower in acute ESAM than NESAM[14,25,75], and similar changes have been noted in response to gut microbiome dysbiosis, which is now well-described in malnutrition[17,18,76,77]. Smith et al.[76], for example, reported significantly lower methionine and cysteine levels in gnotobiotic mice transplanted with the stool microbiome of children with kwashiorkor than those transplanted with stool from their healthy twins. Thus, one hypothetical molecular cascade involves the initiation of aberrant 1-carbon metabolism, exacerbated by gut microbiome dysbiosis, and downstream epigenetic changes that potentiate dysregulation of genes central to expression of the clinical phenotype. The therapeutic corollary of this hypothesis is that augmentation of methyl-flux, through nutritional supplementation of methionine, related 1-carbon derivatives, or associated methyl-donor cofactors (e.g. choline or betaine) before or during acute illness, may be a viable way of abating DNA hypomethylation and mitigating the severity and/or establishment of acute ESAM[78,79], as recently partially suggested by the administration of choline in a mouse model of hepatic steatosis induced by undernutrition[80].

## Methods

**Experimental design.** Demographics of the samples utilized for the present study are given in Supplementary Table 1. DNA samples used in the study were from participants recruited in St. Andrew, Jamaica, and rural sites in five southern Malawi districts. In Jamaica, the study was approved by the Ethics Committees of the UHWI/University of the West Indies Faculty of Medical Sciences. In Malawi, the study was approved by the National Health Science Review Committee (NHSRC) of the Ministry of Health, Government of Malawi. In both countries, written informed consent was obtained from adult participants and the parent or adult guardian of participating children and all ethical regulations regarding human participants were complied with. Permission to use the participant samples for genetic studies at Baylor College of Medicine (BCM) was approved by the Institutional Review Board (IRB) of BCM.

Participants from Jamaica were recruited as part of a long-standing study of genetic susceptibility to SAM being undertaken at the Tropical Metabolism Research Unit (TMRU) of the Caribbean Institute for Health Research (CAIHR) at the University Hospital of the West Indies (UHWI), located in St. Andrew, Jamaica. Participants were either current (prospective) or former (retrospective) patients admitted for in-patient care on the metabolic research ward of TMRU, which serves as a tertiary referral center for children with severe malnutrition from the entire island[81,82]. The principal inclusion criterion for all participants was a

diagnosis of SAM according to the Wellcome Classification[83]; i.e. marasmus (<60% weight-for-age; no edema), marasmic-kwashiorkor (<60% weight-for-age with edema), or kwashiorkor (60–80% weight-for-age with edema). Weight and height of all individuals was measured at the time of recruitment. Prospectively recruited individuals, who were sampled within 8 weeks of their hospital admission, were designated as acute (discharged, DC) samples. Admission records were used to identify former patients living within a 20-mile radius of TMRU. Among these SAM survivors, samples from available adults (older than 17 years) were designated as delayed (DL) samples. The Wellcome classification was used in both prospective and retrospective cohorts in order to facilitate comparability between the groups. Participants with pre-existing chronic illnesses predisposing to potential secondary malnutrition (e.g. HIV seropositive, congenital malformations, cerebral palsy) were excluded from this arm of recruitment.

In Malawi, DNA samples were prospectively obtained between 2013 and 2016 from participants seen at 18 outpatient feeding clinics in rural Malawi[84]. Each child's weight, length, and mid-upper-arm circumference (MUAC) were measured. Children between 6 and 59 months of age, with nutritional edema (indicative of kwashiorkor), and those with a MUAC < 11.5 cm (indicative of marasmus), or both (marasmic-kwashiorkor), were eligible for enrollment. Samples were obtained at the time of initial diagnosis (acute, DC), and participants were categorized as either edematous (kwashiorkor and marasmic-kwashiorkor) or nonedematous (marasmus) for the purposes of the study. MUAC has been noted to be a more sensitive indicator of acute malnutrition than weight-for-age[22,85] and has become the de facto standard-of-care for the diagnosis of NESAM in much of rural Africa. Adhering to the recruitment strategy, edematous and nonedematous participants were matched by age and gender within the study age spectrum. Individuals with overtly compromised or infected oral mucosa were excluded from the study. Acute (fever, diarrhea) and chronic (e.g. HIV) illnesses as well as antibiotic use at the time of recruitment were recorded for each participant.

The mitotic turnover of buccal cells takes anywhere from 5 to 25 days in healthy individuals[86], meaning that incident DNA methylation patterns likely reflect methyl-group turnover in the preceding days to weeks. This is congruent with the concordance of results between the Jamaica and Malawi DC cohorts, despite slight differences in the timings of sample acquisition (Supplementary Data 1). Although SAM classification schemes were different in Malawi and Jamaica, we observed that reclassifying individuals according to the Wellcome criteria in both cohorts strengthened our associations; this suggests that our analysis is more conservative, but more applicable to current recommendations for assessing SAM[87].

**DNA processing and methylation array typing.** In Jamaica, two sets of mouth swabs (ten plastic sterile cotton tip applicators) were used to swab the inner jaw, with five strokes of each jaw taken with each swab[81]. Swabs are then placed in a 15 ml orange top tube with 3 ml of cell lysis solution. In Malawi, buccal epithelium was sampled using the Oragene Discover (OGR-250) DNA collection kit in conjunction with CS-1 and CS-2 sponges (DNA Genotek Inc., Ottawa, Ontario, Canada). The protocol was modified to collect buccal epithelial cells by obtaining ten passes of each inner cheek using five swabs. DNA was extracted according to the manufacturer's instructions.

The quantity of double-stranded DNA (dsDNA) in each sample was assessed using plate fluorescence (PicoGreen, Life Technologies, Grand Island, NY). Bisulfite conversion was performed with the EZ-96 DNA Methylation kit in deep well format (Zymo Research Corp., Irvine, California, USA) on 500 ng of extracted DNA. A random number generator was used to assign samples to plates ahead of methylation array typing. DNA methylation was assessed using Illumina's Infinium HumanMethylation450 Bead Chip array (Illumina, Inc., San Diego, California, USA) following the manufacturer's recommendations for the manual process outlined in the Infinium HD Assay Methylation Protocol Guide.

**Methylation array processing.** IDAT files for all samples, containing raw fluorescence intensity values, were collated and analyzed in R using the ChAMP package (version 1.8.2)[88] to identify low-quality samples. Out of an initial total of 407, 33 samples with >2% failed probes were excluded (all from Jamaica, 19 DL). All remaining 374 samples (Supplementary Table 1) were imported simultaneously using the Methylation Module (version 1.9.0) in Illumina's GenomeStudio software (version 2011.1). Reading all samples together reduced the potential for batch effects in downstream data. Background fluorescence intensity values were subtracted from the remaining IDAT files without control normalization. The resulting fluorescence intensity values were color balanced and quantile normalized using the R package lumi (version 2.22.1)[89]. Probes meeting the following criteria were omitted from subsequent analyses: internal controls ($N = 65$), detection $P$ value < 0.0001 in >10% of samples ($N = 860$), bead counts <3 in 5% of samples ($N = 665$), sex chromosomes ($N = 11,288$), cross-reactive ($N = 29,895$), and containing SNPs with a minor allele frequency (MAF) >0.01 in African populations at, or within 5 bp of, the single base extension (SBE) site ($N = 16,918$ and $N = 5386$, respectively)[90]. A total of 420,500 probes remained for differential methylation analysis. All reported genome coordinates were referenced to hg19.

**Methylation analysis.** Methylation estimates resulting from the lumi preprocessing steps in the form of $M$ values were normalized using the qqnorm function in

R (v3.4.1). Processing with the COMBAT module of the ChAMP suite[88] showed no evidence for systematic biases on the basis of chip or batch using single value decomposition (Supplementary Fig. 7).

Resulting $M$ values were fit to a linear regression model to identify single probes with evidence of differential methylation between ESAM and NESAM samples: $M_i = Ca + Yb + e$, where $M$ = normalized methylation value at probe $i$, Ca = matrix of covariates, Yb = matrix of case (ESAM) or control (NESAM) status, and $e$ = random error. Acute (DC) samples were analyzed separately from adult recovered (delayed; DL) samples. Principal component analysis (PCA) using methylation values for all probes demonstrated no systematic global differences between DC samples from Jamaica and Malawi (Supplementary Fig. 8), and these were combined to increase the available sample size and mitigate age differences between ESAM and NESAM children observed in each country. In the subsequent analysis, age, gender, and the first principal component (PC1), which accounted for 20.6% of the variance in the data, were utilized as covariates. Preference was given to PC1 as a covariate over geographic sampling location as it improved the genomic inflation factor (from 1.24 to 1.16) and thus likely encompassed other sample variation in addition to location; inclusion of additional principal components was found to increase the genomic inflation factor (Supplementary Table 3). For the DL analysis, there were no systematic sample differences noted on PCA (Supplementary Fig. 8); gender and age were included as covariates.

To provide evidence that differentially methylated sites were consistent across countries, DC individuals from each country group were also assessed separately. In the Jamaican DC cohort, ESAM children were significantly younger than those with NESAM, and vice versa in the Malawian DC cohort (Supplementary Table 1). To account for this confounding, age was omitted as a covariate in the country-specific methylation analysis, and instead, 2459 probes shown to be sensitive to age in buccal epithelial cells[91] were removed. Gender was utilized as a covariate under the same linear regression model used for the joint analysis.

We validated our results from the 450 K beadchip by performing capture-based bisulfite sequencing (SeqCap Epi CpGiant Enrichment Kit, Roche NimbleGen, Inc., Madison, Wisconsin, USA)[92] on three samples from Jamaica that also had array-based methylation. Overall correlation between array and sequencing at overlapping high-quality sites was 0.95 (Pearson's $r$; Supplementary Fig. 9).

**Methylation clustering using CpG correlation.** Methylation at neighboring CpG loci tends to be correlated[31]; therefore, in order to reduce false positive associations at individual sites, adjacent CpG sites were binned into clusters on the basis of their magnitude and direction of association. Single probes with effect sizes [beta$_{slope}$] between −0.05 and +0.05 were removed to reduce ambiguity of the direction of effect at the margins of association, which could change cluster assignment. Of the 166 Bonferroni DMCs, 19 would have included >2 of these marginal probes. The remaining probes were binned into nonoverlapping clusters in which the distance between adjacent probes was at most 10 kb, and all binned probes shared the same direction of effect (Supplementary Fig. 10). This resulted in 135,503 and 173,635 clusters for the DC and DL analyses, respectively. The regression analysis described above was then repeated using the mean $M$ values of all probes within a cluster as the dependent variable. The resulting DC clusters encompassed anywhere from 1 to 43 probes, with a median size of 1.7 kb (range 4 bp to 41.5 kb for clusters with >2 probes).

**Cellular composition of samples.** To confirm that our DNA samples were primarily derived from buccal epithelial cells in our dataset of 309 DC and 65 DL samples, we isolated available CpG sites that were needed for the online Horvath DNA methylation age calculator (accessed September 21, 2017)[93], which provides probabilities for potential tissues of origin based on tissue-specific methylation patterns. According to the tissue prediction algorithm, 97.4% of the 309 DC samples were primarily buccal (epithelial) in origin (median probability of predicted buccal samples: 0.84), and this proportion was similar in ESAM (162/164 (98.8%)) and NESAM 139/145 (95.9%) (Supplementary Fig. 11). Similarly, a secondary method of cell type heterogeneity assessment, EpiDISH[94] used in combination with EpiFibIC reference data[95] and its robust partial correlations (RPC) method, predicted epithelial as the major cell fraction in 91.6% (283/309) of acutely malnourished children, with no difference between ESAM and NESAM (chi-squared test, $P = 0.12$). This was consistent with anecdotal reports from field recruitment that many of the acutely malnourished children were also dehydrated and had insufficient saliva for collection. We also performed capture methylation sequencing of buccal and peripheral blood DNA sampled <3 days apart from the same three individuals; mean Pearson correlation within-individuals (0.84) was less than between the same tissues from different individuals (0.96). Pearson correlation between buccal and blood methylation from bisulfite sequencing data was assessed at 116/157 Bonferroni-significant single sites that were captured at any coverage in both tissues and had nonzero variance between samples. Among DL adult samples, 58.5% were estimated to be primarily buccal epithelial (median probability of predicated buccal samples: 0.31), while the remaining samples were predicted to have originated from saliva (40.0%) or whole blood (1.5%); however, this proportion was not significantly different between former ESAM and NESAM. Similarly, our secondary method predicted epithelial as the major cell fraction in 27.7% (18/65) of adult samples, without differences in ESAM and NESAM (chi-squared test, $P = 0.82$). On the basis of these results we concluded that the primary

comparisons between ESAM and NESAM in each group are not significantly impacted by cellular composition, although this may have a greater impact on comparisons between DC and DL groups.

Pearson correlation was assessed between blood and buccal at CpG loci that passed the Bonferroni threshold of significance in our single site analysis and that were captured by bisulfite sequencing, according to the approach described in ref. [92], in the three individuals with paired buccal and blood samples (Supplementary Data 9).

**Gene ontology and pathway enrichment analysis.** RefSeq genes overlapping or within 10 kb of Bonferroni-significant clusters were taken as candidate genes. ConsensusPathDB's over-representation analysis (accessed June 29, 2017)[96] was used to identify enriched gene ontologies (hypergeometric test, FDR < 0.01) associated with these candidates. RefSeq genes overlapping or within 10 kb of the CpG loci targeted by the array were used as background. Gene ontology analysis included all levels with default parameters.

**GWAS study enrichment score.** The NHGRI-EBI catalog of published genome-wide association studies (GWAS)[33] includes GWAS hits labeled with terms from Experimental Factor Ontology (EFO; https://www.ebi.ac.uk/efo/). We first considered EFO labels of GWAS hits that were mapped to one of our top differentially methylated genes. Each pair of an EFO and gene was then given a score based on the number of SNPs mapped to the gene in separate GWAS hit records labeled with the EFO term. The final scores were then projected onto the differentially methylated genes, focusing on EFOs represented by more than one gene. To assess the relative likelihood of individual EFOs, we generated a list of 237 genes (replicating the number of genes associated with our Bonferroni-significant loci) randomly selected from the list of 20,622 genes linked to 450 K array probes. This random selection was repeated 10,000 times and the GWAS enrichment score was evaluated in each instance.

**Phenotype-gene ontology scores.** Human Phenotype Ontology terms[34,35] associated with kwashiorkor were used as ESAM-associated phenotypes, with the exception of global developmental delay, which was considered to be an overly broad term for development in acute malnutrition (motor delay was retained). From this list, genes associated with each term were obtained from the HPO site (build #129; accessed June 2017). These genes were then run through ConsensusPathDB as described above, using the same set of background genes, to derive a set of kwashiorkor-phenotype gene ontologies (KGO) with FDR < 0.01.

To quantify the overlap between study gene-ontologies (SGO) (Supplementary Fig. 4; Supplementary Data 5) and phenotypes associated with kwashiorkor, we derived a weighted enrichment score. First, enriched SGOs and KGOs were ranked by $P$ value, such that each SGO ontology was given a value equal to the $P$ value quartile in which it fell (i.e. 1, 0.75, 0.5, 0.25) multiplied by its scaled quartile place in the KGO list. Therefore, SGOs that are not found in the KGO list receive a score of 0 by default, and SGOs that are found in the KGO list and in the top quartile of both lists would have a score of 1 (the highest score). Then, for each kwashiorkor HPO term, we identified the genes associated with that term and calculated the proportion of those genes that mapped to each SGO. The final score for each SGO −phenotype pair was then derived by taking the arithmetic mean of the two scores ((quartile score + proportion score)/2); this final score ranged from 0 to 1.

**DNA/RNA dual collection and processing.** Twenty-eight additional Malawian DC buccal samples had both DNA and RNA isolation, and these were used for gene expression-methylation correlation analysis. DNA was extracted as outlined above, and methylation was assessed using the Illumina EPIC methylation kit in 24 samples following procedures as outlined for the main dataset. The Oragene RNA (RE-100) RNA collection kit was used in conjunction with CS-2 sponges (DNA Genotek Inc., Ottawa, Ontario, Canada) with modification, to collect buccal epithelial cells from multiple cheek swabs. RNA was extracted according to the manufacturer's instructions. Gene expression analysis was undertaken for 24 samples using the Affymetrix GeneChip human Clariom S assay (Thermo Fisher Scientific, Waltham, Massachusetts, USA) following the recommendations of the GeneChip WT PLUS Reagent Kit—Target Preparation for GeneChip Whole Transcript (WT) Expression Arrays manual.

**Expression analysis.** Gene expression levels in the form of fluorescence intensity values stored in CEL files were background corrected and normalized on Affymetrix's Expression Console software (build 1.4.1.46, Thermo Fisher Scientific, Waltham, Massachusetts, USA) using the SST-RMA algorithm. Probe sets with expression levels twofold above the mean expression levels of the antigenomic control probes, which are designed to be incompatible with any human genome sequence, were considered as expressed.

**Methylation-expression correlation.** Paired DNA methylation and expression data were available for 20 out of 28 samples (remaining samples failed quality control for either methylation or gene expression, or both), who did not differ from the previous Malawian cohort with regard to age, gender, and ESAM/NESAM

proportions. Spearman correlation coefficients were calculated between the methylation beta- and $M$ values for each CpG site and the expression value for each gene within 10 kb of the differentially methylated cluster into which the site was binned. Only genes with expression levels twofold above the antigenomic background were assessed. This resulted in 151 CpG-gene expression comparisons, consisting of 146 unique CpG loci and 25 unique expressed gene transcripts, corresponding to 25 genes. A post-hoc power analysis using G*Power (version 3.1.9.2)[97], set to exact test family and correlation: bivariate normal mode with $\alpha = 0.05$ and $N = 20$, revealed that, dependent on the minimum ($|rho| = 0.45$) or maximum ($|rho| = 0.65$) coefficient associated with significant correlations, our analysis has a power of 53% or 91%, respectively.

**Genotyping**. DNA samples from the broader Jamaica cohort[81,82] were genotyped on two separate genotyping platforms: HumanOmni1-Quad BeadChip (Illumina, San Diego, CA) and HumanOmni2.5 BeadChip (Illumina, San Diego, CA). Quality control was performed for SNPs and samples in each phase individually and then on the merged set. GenomeStudio software (Illumina) was used to interpret normalized fluorescence intensities as genotypes. SNPs with more than 2% missing data (genotyping efficiency <0.98), SNPs out of Hardy−Weinberg equilibrium (exact test, $P < 1 \times 10^{-3}$), or with MAF < 0.05 were removed. Samples with more than 2% missing data (genotyping efficiency <0.98), or with evidence of excessive inbreeding (inbreeding coefficient, F > 0.1) were removed. Pairs of samples with excess allele sharing suggestive of close familial relationships (parent−offspring, siblings; Pi_HAT > 0.1) were identified using identity-by-descent (IBD) analyses, and in such cases one member of the pair was removed from analysis ($N = 22$). After merging of datasets and sample quality control (QC), 384 samples and 450,012 autosomal SNPs remained; 138 samples had both adequate genotyping and methylation data for analysis after QC (53 ESAM-DC, 37 NESAM-DC, 23 ESAM-DL, and 25 NESAM-DL).

**Population ancestry**. Genotyped individuals were evaluated for shared ancestry using the multidimensional scaling (MDS) method found in PLINK[98]. A subset of 100,330 independent SNPs ($r^2 < 0.1$; window size 50 SNPs per nonoverlapping window) was used for analysis. ESAM and NESAM samples clustered together on the first and second MDS dimensions and showed close ancestry with African populations from the 1000 Genomes Project Phase 3, particularly the Yoruba of West Africa (YRI) (Supplementary Fig. 12). MDS values were used as covariates of ancestry in meQTL analyses.

***Cis*-acting methylation quantitative trait loci (meQTLs)**. Identification of meQTLs was undertaken using the R package MatrixEQTL[99] (version 2.1.1). Initially a linear regression model (Probe value = Covariates (age, sex, first two MDS components) + SNP (0, 1, 2 minor alleles)) was used for all SNPs within 10 kb of a CpG. Analyses were performed separately for DC and DL samples. To test for nutrition-dependent specific effects, meQTLs significant in DC samples ($P \leq 0.05$) that also overlapped FDR-significant differentially methylated clusters were selected. Analyses were performed using a separate linear regression model: Probe value = Covariates (Age, sex, first two MDS components) + SNP (0, 1, 2) + SAM type (ESAM or NESAM) + SNP × SAM type. MeQTLs with a significant ($t$ test, $P < 0.05$) SNP × SAM type interaction were then reported separately for ESAM and NESAM (Supplementary Data 6). Significant meQTLs, with DC-specific ESAM-NESAM interactions as well as those without any evidence of ESAM-NESAM interactions, were compared to meQTLs identified by Gaunt et al.[60] in cord and peripheral blood at different life stages. Counts of meQTL overlap at any life stage were compared using Fisher's exact test.

**meQTL overlap with transcription factor binding motifs**. MotifBreakR[57] (version 1.6.0) was applied to identify transcription factor binding motifs described by Jolma et al[56]. that coincided with SNPs from significant meQTLs. The function's default parameters were changed to allow a maximum match-calling $P$ value threshold of $1 \times 10^{-4}$, filter by $P$ value, use of the information content (ic) method, and to show results with neutral effects on transcription factor binding likelihood. SNPs with an absolute score difference >1.5 between reference and alternate allele were considered to indicate a strong likelihood of altering transcription factor binding (Supplementary Data 7 and 8).

**Reporting summary**. Further information on research design is available in the Nature Research Reporting Summary linked to this article.

## Data availability

Methylation datasets generated during the current study are available on the Gene Expression Omnibus (GEO) database under the accession number GSE112893. The source data underlying Fig. 3a, c, Supplementary Fig. 5a–d, and Supplementary Table 1 are provided as a Source Data file. All other relevant data supporting the key findings of this study are available within the article and its Supplementary Information files or from the corresponding author upon reasonable request. A reporting summary for this Article is available as a Supplementary Information file.

## Code availability

All analyses were carried out in the programming language R (versions 3.2.0 and above) as described in the methods. Software versions and relevant parameters are included in the corresponding methods sections. Scripts are available from the authors upon request.

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

## Acknowledgements

The authors would like to thank Rob Waterland, Michael Wangler, and Farook Jahoor for valuable advice and insight. Thank you also to Dr. Jenny van Dongen, who furnished summary statistics for buccal-derived meQTL from the Netherlands Twin Study. We would also like to acknowledge the efforts of Prof. Colin McKenzie, who died prior to publication of the manuscript, in developing and advocating for the molecular study of ESAM. This research was supported by Clinical Scientist Development Award from the Doris Duke Charitable Foundation (Grant #: 2013096) and USDA, ARS cooperative agreement (58-3092-5-001), both to N.A.H. K.V.S. and N.C.L. were supported by Award Numbers T32GM008307 and 5 T32GM008231, respectively, from the National Institute of General Medical Sciences. The content is solely the responsibility of the authors and does not necessarily represent the official views of the National Institutes of Health.

## Author contributions

J.W.B., I.T., M.J.M., C.A.M., M.E.R., C.T.-B., and N.A.H. designed the study; O.B., C.T.-B., I.T., T.M., and M.J.M. recruited participants, obtained informed consent, and collected samples; S.H., K.M., R.S., N.H., X.W. molecularly characterized samples; A.J., S.S., N.C.L., L.Z., Y.G., T.M., K.V.S., J.W.B., and N.A.H. conducted data analyses; K.V.S., S.S., A.J. and N.A.H. wrote the paper. All authors read, edited, and approved the manuscript.

## Competing interests

J.W.B. and X.W. are fulltime employees of Illumina Inc., but all work relevant to the current study was performed whilst they were employees of Baylor College of Medicine. The remaining authors have no competing interests to declare.
