## [Peer Review File · Nature Communications]

Reviewers' comments:

Reviewer #1 (Remarks to the Author):

In the study 'DNA hypomethylation in acute kwashiorkor provides a molecular link to aberrant 1-carbon metabolism in severe malnutrition', Schulze et al compare genome-wide DNA methylation between individuals with two forms of malnutrition, edematous severe acute malnutrition (ESAM) and non-edematous severe acute malnutrition (NESAM), identify genomic regions differentially methylated between ESAM and NESAM, and perform enrichment analysis to illustrate associations between DNA hypomethylation and ESAM pathology.

I appreciate the authors' effort in identifying ESAM-specific epigenetic features and agree that epigenetic modifications such as DNA methylation are important in mediating the influence of nutritional status on ESAM pathology, my major concern is that this study does not provide direct evidence, as its title implies, to support the casual relationship between dysregulated one-carbon metabolism and DNA hypomethylation in ESAM patients. This study is mainly a comparative analysis showing that compared to NESAM, ESAM patients display DNA hypomethylation enriched at loci somehow associated with malnutrition-related phenotypes. Given that DNA methylation is regulated by many factors including activities of DNA methyltransferases and demethylases and availability of substrates and cofactors used by these enzymes, such a comparison is not sufficient for establishing a molecular link. Thus the conclusions of the manuscript need to be tempered. Besides the lack of direct evidence supporting molecular link between 1C metabolism and DNA methylation in ESAM patients, I have other concerns regarding the strength of evidence for association between DNA hypomethylation and ESAM pathology, the statistical methods used, and the presentation of results. These major points are listed below:

1. As I mentioned, the evidence is not strong enough to support the association between DNA hypomethylation and dysregulated nutrition and metabolism in ESAM patients. To corroborate the association between the differentially methylated clusters (DMCs) and ESAM pathology, the authors use two lines of evidence, both involving association with terms very weakly related to malnutrition and metabolism. For the GWAS analysis, a lot of associated terms have little to do with malnutrition. The second evidence is only based on overlap between KGOs (i.e. gene ontologies enriched in genes related to kwashiorkor in HPO) and gene ontologies enriched in the DMCs, but it is still very likely that there is actually no overlap between the two groups of genes.
2. The comparison between EFO terms related to genes associated with the DMCs and those related to randomly selected genes is confusing. My understanding is that the frequency of EFOs in the randomly selected genes (Fig S1) gives an approximation of a null distribution for association between EFOs and genes in the GWAS catalog, thus it is straightforward to estimate the significance of enrichment of each EFO in the DMCs based on this null distribution. Alternatively this could also be done easily with a hypergeometric test or so. However, in the manuscript, there is only one statement that is very confusing: 'less than 6% of these random gene permutations resulted in as many or more (≥ 33) GWAS catalog genes being mapped to at least as many (≥ 32) associated EFO terms as our results'. It is not clear to me what this means.
3. The correlations between DNA methylation and gene expression, as shown in Figure 3, are in general very weak, and whether the correlations tend to be positive or negative for different genomic elements is difficult to read from the figure since almost all medians are close to zero. It would be helpful to add the $y=0$ line to this figure and perform some statistical hypothesis testing (e.g. a Wilcoxon's signed rank test) to evaluate the deviation of the medians from zero. It is also necessary to evaluate the significance of each correlation coefficient using more rigorous statistical tests instead of

absolute values of correlation coefficients.

4. Figure 5 shows some genes associated with DMCs in ESAM. Since the correlations between DNA methylation and gene expression are very subtle, it is necessary to further show that the changes in DNA methylation are indeed coupled to changes in expression levels of these genes. Moreover, the genome browser tracks in this figure are not very informative. I recommend removing them or moving to supplementary figures.

5. It is not clear how the list of 34 phenotypes associated with 'kwashiorkor' was generated based on HPO since searching for the term 'kwashiorkor' in HPO resulted in nothing. This needs more details to make sure that the readers can reproduce the results.

6. Several terms need clear explanation. For instance, the slopes from meQTL analysis (Fig 6A) are not explained in the manuscript. My guess is that they are the coefficients for the SNP term in the linear regression model but this needs clarification.

7. The authors have predicted the effects of nutrition-sensitive meQTLs on TF-binding and found that they are likely to alter the likelihood of TF-binding. This is interesting but a lot of necessary details are missing. It is unclear that what TFs are included in the prediction and whether other meQTLs (i.e. those nutrition-insensitive ones) are also able to change the TF binding affinities with a high chance.

8. On page 13: 'In fact, 40% of our differentially methylated sites showed strong correlation (Pearson $r \geq 0.7$) between buccal and blood DNA methylation (Methods)'. It is necessary to provide figures or datasets to support this statement.

Minor:

Font sizes in Fig 4 are too small to read.

Reviewer #2 (Remarks to the Author):

Schulze et al. analyzed genome wide methylation pattern in (a) severe acute childhood malnutrition (edematous SAM or ESAM) and in non-edematous SAM (NESAM). Transmethylation of 1-carbon substrates has been reported to be slower in acute, but not recovered, ESAM. So that DNA methylation changes could lead to the differences in acute pathogenesis between the two forms. Genome-wide DNA methylation was analyzed in 309 SAM children from two independent cohorts. ESAM, compared to NESAM, was characterized by significant hypomethylation at 99% of differentially methylated loci. Interestingly, adults recovered from SAM displayed no significant differences in methylation pattern between ESAM and NESAM. The authors claimed that gene expression and methylation at hypomethylated loci demonstrate both positive and negative correlations, so that a complex transcriptional response to the acute disturbance can be assumed. The authors showed that genes at hypomethylated loci also lit up in the Human Phenotype Ontology sub-phenotypes of kwashiorkor. They found enrichment for genes linked to nutrition and metabolism. Nearby genetic variation seems to influence methylation at these loci.

Major:

There are few previous studies pertaining to methylation pattern in severe malnutrition (e.g. references 68 and 69: C. J. Peter et al., DNA Methylation Signatures of Early Childhood Malnutrition Associated With Impairments in Attention and Cognition. *Biol Psychiatry* 80, 765-774 (2016); A. Sheppard et al., Molecular Evidence for Differential Long-term Outcomes of Early Life Severe Acute

Malnutrition. EBioMedicine 18, 274-280 (2017)) These should be described in more detail in the introduction.

Did the authors confirm the methylation patterns with an independent method?

Minor

Abstract: it should be mentioned that buccal cells were used for the analyses.

Abstract: it should be mentioned that Illumina arrays were used.

Supplementary Table S4: cg03637179 - p-value is not below 0.05, please remove color.

Reviewer #3 (Remarks to the Author):

In "DNA Hypomethylation in Acute Kwashiorkor Provides a Molecular Link to Aberrant 1-Carbon Metabolism in Severe Malnutrition" by Schulze et. al., the authors compare the DNA methylation profiles of buccal epithelial cells from children with edematous and non-edematous SAM in two countries, as well as adults who recovered from SAM. The authors seek to link the clinical and metabolic differences observed between the two phenotypes with underlying epigenetic mechanisms. They demonstrate that almost all differentially methylated CpGs between the two cohorts are hypomethylated in ESAM, that nearby genes are associated with kwashiorkor-linked symptoms, and that genetic variation may influence the change in methylation. In contrast, there is no difference in DNA methylation between the subtypes in recovered adults.

Severe malnutrition is a critical issue worldwide, and this paper is both important and would be of interest to a broad readership. However, it is unclear whether the methylation effect size is biologically meaningful, and the last two results sections are difficult to parse without a schematic. However, although I have some critiques, I would recommend this paper for publication.

Major comments:

The mean absolute difference in methylation beta-values even for significant CpGs is very small (0.054). Is this biologically meaningful? The authors should consider introducing a threshold to identify sites with the strongest difference between groups.

It is unclear exactly how the authors performed the intra-individual comparison of CpG methylation to gene expression. They state that they compare each CpG to the expression of all genes within a 10kb region, but Fig 3B is divided by genic location. It is conceivable that a CpG in the promoter of one gene does not influence the expression of a nearby gene, which may explain why so many of the comparisons in Fig3B are not strong (rather than a "complex transcriptional response"). A table of the number and location of DMCs corresponding to each gene and vice versa would help resolve this. Additionally, the authors should clarify whether CpGs over different genic features have different effect sizes.

On Line 248, the authors state that the meQTLs are predicted to impact TF binding. Which TFs are implicated?

The authors do not speculate as to why slower carbon metabolism would specifically impact genes implicated in the presentation of ESAM. Are these genes poised in some way that makes them more sensitive to carbon metabolic changes than other sets of genes? The correlation between buccal cells and blood suggests that this is agnostic to tissue.

More justification and analysis should be provided to address heterogeneous cell types in buccal samples.

Minor comments:

- Figure 3A displays the results only for Malawian children. Why are the Jamaican children not

included?

- A schematic, including the number of ontology terms and genes included in each dataset, would assist in the interpretation of the section Differentially methylated loci converge upon genes related to nutrition and metabolism.
- Line 157: Is the number of EFO terms informative? For instance, one gene with a very large number of linked EFOs, or several related EFOs, could skew this analysis.
- Line 251: What does this study investigate?
- Line 506: Is there a percentage associated with “primarily”? Does it differ between subtypes? Were the samples that were predicted to originate from other sites excluded?

Reviewer #4 (Remarks to the Author):

This is a thorough, fascinating study on the potential molecular mechanisms that may be implicated in the various physiological differences between the edematous and non-edematous forms of severe acute malnutrition. Given there is increasing awareness of the true prevalence of kwashiorkor (and that it is likely underestimated globally), combined with the fact that it remains unknown why kwashiorkor develops in some households / communities and not others, this is a relevant study of public health significance. Strengths of the study are many, for example, by obtaining DNA methylation samples from the patients during the acute phase of the illness, along with the integration of gene expression data.

At the outset I will be transparent that the focus of my review has been on aspects specifically related to SAM, one-carbon metabolism and the public health implications. I defer to other referees with greater bioinformatics expertise for more insight on the methods.

Whilst I do think this is a novel study of interest to both epigeneticists and public health nutritionists, I have made some specific points below. In summary, my main concern is that the hypothesis and associated discussion sections appear to rest on a single study that does not support what you claim in the narrative. There is no evidence of decreased methionine transmethylation in children with edematous malnutrition in the referenced study. Therefore, I could suggest you place more prominence in your introduction and discussion on the literature showing reduced methionine concentrations in edematous malnutrition compared to marasmic patients. This could still feed into your rationale for investigating DNA methylation and would provide a stronger evidence base.

I found that in several places your language could benefit from minor editing to be more conservative in your statements, especially those involving causality. There were also some further instances where your references did not back up your arguments. Finally, I suggest that you edit your conclusion about future public health / clinical interventions for treatment of kwashiorkor, which I feel are currently premature at this stage.

I do think, however, that this study makes an important contribution to the understanding of how kwashiorkor is characterised at the molecular level, and trust that the below comments are relatively easy to address.

Specific points

Introduction

Line 47: In many contexts where NESAM is treated, weight loss is often acute rather than chronic.

ESAM is generally described as the presence of nutritional (or bilateral) edema to distinguish it from other (unilateral) forms. I suggest clarifying.

Line 54: Consider removing 'innate', as this suggests the biochemical and physiological differences between the two forms of SAM are consistent between populations. Yet, as you have correctly mentioned in the same line, our understanding of the conditions are based on fragmented data that are far from cohesive.

Line 58: Again, I suggest you soften the language by removing 'clear'. When we look at the hardest of clinical outcomes, mortality, it is actually not clear. In some countries kwashiorkor is associated with increased mortality compared to marasmus, but this is not a consistent pattern and an area of continued research.

Line 59: I found the purpose of talking about the unchanged nature of SAM treatment for both conditions unclear here. Are you suggesting there should be different approaches? If so, the case should be made. In community management of acute malnutrition using ready-to-use therapeutic food there is very strong evidence that grades 1+ and 2+ of kwashiorkor can be treated successfully in outpatient settings, using the same protocol as for marasmus.

Lines 60-62: As per comment above, it needs to be a bit clearer whether this section is general background or setting up your study rationale. It feels a bit out of place at present.

Line 63-64: Reference 24 is about protein turnover more generally and makes no claim to the specific components of 1-carbon metabolism. I suggest removing it.

Line 64-65: You mention that in reference 26 there is a slower transmethylation (TM) of methionine in ESAM children during the steady-state re-feeding stage. Please check this. The authors state in the results, and again in the discussion, that 'there were no significant differences in homocysteine remethylation, methionine transsulfuration, or methionine transmethylation between the groups at clinical phase 1'. Please also change the abstract (line 26) accordingly.

This is a critical point for your paper as your rationale is centred on the hypothesis that a reduced transmethylation rate that could potentially lead to lower DNA methylation. The reference does not support this. Indeed, it actually suggests that despite lower protein breakdown the transmethylation rate is maintained.

Lines 70-71: Given the study (ref 26) did not show a significant difference in TM between the groups, I feel your causal statement about reversal of biochemical changes from dietary recovery is too strong.

Lines 71-72: "The phenotypic differences that characterize the two forms of SAM are only evident during acute malnutrition". Wouldn't your reference 69 (Sheppard et al. 2018) suggest otherwise?

Line 76-77: Given a number of 2014 studies (e.g. PMID 24980666, 24517147), alongside Sheppard et al. 2018 do suggest phenotypic differences in SAM-survivors by NESAM/ESAM, I was curious why you hypothesise that methylation differences would have dissipated by recovery? You mention one of these studies in line 316. It therefore seems you are basing your whole hypothesis on reference 26, which as mentioned above, does not seem to give sufficient evidence (of TM differences), even during the acute phase.

Results

I felt this section could benefit from being cut down if possible. Perhaps some of the details on the individual genes in lines 183-212 could be integrated in the discussion to ease the narrative flow?

A lot of methodological detail / justification was also included here, which could be edited down given you have a detailed methods section.

Discussion

Line 268: It is difficult for the reader to understand which types of studies you are referring to, so I suggest cutting the sentence.

Lines 272: This will need editing if you agree with my earlier comment that reference 26 does not report slower TM in ESAM.

Lines 279-280: Your reference 58 does not support your statement, and this will in any case very much depend on the genomic region. The Lowe et al. reference is comparing blood and buccal, and if anything emphasises tissue DMRs, not cross-tissue methylation similarities.

Line 290: delete 'acute' as SAM is acute by definition. Same in line 29.

Line 291: Soften the language here – to say a transcriptional profile is consistent with results from a referenced study suggests to the readers the studies are looking at the same topics and have similar results, but reference 59 is not about your topic specifically. Here you are simply stating that methylation-transcription dynamics are complicated, which you already mentioned in line 290 (so you could delete this sentence).

Line 302-305: I find this a little confusing since your hypothesis is on one-carbon metabolic components driving transmethylation rates, rather than energy sources (and dietary intakes for ESAM and NESAM are very often the same). Furthermore, the thrifty genotype hypothesis would presumably extend to ESAM as well as NESAM? It may be easiest to delete this section to avoid readers being distracted from your core messages, as you already have a lot of good content in the discussion.

Line 338: This is very interesting. Please reference the studies you are referring to regarding different dietary intakes for those developing kwashiorkor.

Line 341: Throughout the discussion, and particularly here and lines 351-354, you have been strong in suggesting a putative causal association between aberrant 1-carbon metabolism, hypomethylation and the type of SAM. I suggest editing the language to be more cautious in reflecting the real possibility of reverse causation, which I do not feel you ever fully address. The intricate feedback mechanisms involved in 1-carbon metabolism suggest that changes in methionine supply are very well buffered, and may not actually affect DNA methylation (in addition to reference 26, see Nijhout et al. 2006, PMID 1799881, which would contest your hypothesis).

Lines 345-6: Yes, the evidence base for lower methionine concentrations in ESAM is more widely replicated than your reference to altered TM rates in ref 26. I suggest you re-frame your rationale using these studies.

Line 356: Re-phrase, as choline and betaine are methyl donors. Relevant cofactors in 1-carbon metabolism, leading to effective methionine transmethylation and remethylation would include B6, B12, B2.

Line 356-7: I do not think your suggestion of methionine supplementation is justified by your study at this stage. The focus of your study is on the potential molecular etiology of ESAM compared to NESAM. The reality in public health nutrition programming is that there is a very effective community treatment for kwashiorkor once detected. Are you suggesting that treatment is inadequate and that supplementation with methionine would work above and beyond current treatment protocols? If so, please justify. Furthermore, your reference 77 does not support your hypothesis – it was looking at a way to improve glutathione synthesis and did not test whether it was a viable way of preventing DNA hypomethylation. At most you can advocate for further research into epigenetic effects of methionine supplementation in patients with ESAM.

Philip T James.

Reviewers' comments:

Reviewer #1 (Remarks to the Author):

In the study 'DNA hypomethylation in acute kwashiorkor provides a molecular link to aberrant 1-carbon metabolism in severe malnutrition', Schulze et al compare genome-wide DNA methylation between individuals with two forms of malnutrition, edematous severe acute malnutrition (ESAM) and non-edematous severe acute malnutrition (NESAM), identify genomic regions differentially methylated between ESAM and NESAM, and perform enrichment analysis to illustrate associations between DNA hypomethylation and ESAM pathology.

I appreciate the authors' effort in identifying ESAM-specific epigenetic features and agree that epigenetic modifications such as DNA methylation are important in mediating the influence of nutritional status on ESAM pathology, my major concern is that this study does not provide direct evidence, as its title implies, to support the casual relationship between dysregulated one-carbon metabolism and DNA hypomethylation in ESAM patients.

We thank the reviewer for these considerations. We have edited the title to temper any expectation of evidence for a causal relationship between dysregulated one-carbon metabolism and DNA hypomethylation in children with ESAM (page 1, line 1).

This study is mainly a comparative analysis showing that compared to NESAM, ESAM patients display DNA hypomethylation enriched at loci somehow associated with malnutrition-related phenotypes. Given that DNA methylation is regulated by many factors including activities of DNA methyltransferases and demethylases and availability of substrates and cofactors used by these enzymes, such a comparison is not sufficient for establishing a molecular link. Thus the conclusions of the manuscript need to be tempered.

We certainly understand the concerns of the reviewer given the vast number of factors that are important to DNA methylation. In accordance, we have tempered speculations as to the molecular cause of our observations (page 17, lines 365-367). We note, however, that 1-carbon (1C) metabolism is a major factor in DNA methylation and there is a body of evidence supporting aberrant 1C metabolism in ESAM that could account almost completely for our primary observations. It is difficult to envisage an alternate, but still parsimonious, hypothesis that completely separates our observations of hypomethylation from the biochemical studies showing slower methyl-flux/1C metabolism in ESAM. Therefore, our best working explanation for the data remains that of a link between aberrant 1C metabolism (irrespective of the underlying cause) and the observed DNA hypomethylation (page 17, lines 367-370; 384-387).

Besides the lack of direct evidence supporting molecular link between 1C metabolism and DNA methylation in ESAM patients, I have other concerns regarding the strength of evidence for association between DNA hypomethylation and ESAM pathology, the statistical methods used, and the presentation of results. These major points are listed below:

1. As I mentioned, the evidence is not strong enough to support the association between DNA hypomethylation and dysregulated nutrition and metabolism in ESAM patients. To corroborate the association between the differentially methylated clusters (DMCs) and ESAM pathology, the authors use two lines of evidence, both involving association with terms very weakly related to

malnutrition and metabolism. For the GWAS analysis, a lot of associated terms have little to do with malnutrition. The second evidence is only based on overlap between KGOs (i.e. gene ontologies enriched in genes related to kwashiorkor in HPO) and gene ontologies enriched in the DMCs, but it is still very likely that there is actually no overlap between the two groups of genes.

We understand the desire for direct, observational evidence for the link between hypomethylation and ESAM pathophysiology. It is, however, important to note that this study is the first molecular exploration of ESAM; there are no GWAS of SAM (or ESAM) and there are few, if any, gene-pathways or classes of genes that are definitively associated with ESAM (page 4, line 68; page 8, lines 167-169). For the few genes that have been implicated in SAM from model organism studies, we find good overlap with our hypomethylated study genes (page 10, lines 208-222). To gain a collective view of these genes, however, we had to rely upon broad proxies of genes in GWAS or the HPO to make inferences about the interrelationships between DNA hypomethylation, metabolism, and ESAM pathology. Among these, diabetes, inflammation, and metabolism, and anthropometry, emerged agnostically from our interrogation, and these broad classes are, from physiological studies, known to be fundamental to malnutrition. We thus believe that our results provide reasonable evidence in support of an association (though not causation) between DNA hypomethylation and genes that could plausibly play a role in the dysregulated nutrition of ESAM.

2. The comparison between EFO terms related to genes associated with the DMCs and those related to randomly selected genes is confusing. My understanding is that the frequency of EFOs in the randomly selected genes (Fig S1) gives an approximation of a null distribution for association between EFOs and genes in the GWAS catalog, thus it is straightforward to estimate the significance of enrichment of each EFO in the DMCs based on this null distribution. Alternatively this could also be done easily with a hypergeometric test or so. However, in the manuscript, there is only one statement that is very confusing: ‘less than 6% of these random gene permutations resulted in as many or more (≥ 33) GWAS catalog genes being mapped to at least as many (≥ 32) associated EFO terms as our results’. It is not clear to me what this means.

Thank you for pointing out the need for clarification. The data shown in Figure S1 is an approximation of a null distribution for associations between EFOs and genes in the GWAS catalog. The quoted statement was intended to convey that for each random experiment we counted the number of EFO terms that were represented at least twice by our random gene lists. We observed that 94% of generated random sets (using 237 genes) represented less than 33 EFOs, i.e. it is unlikely to see that many EFOs represented by a set of random 237 genes. We have rephrased this statement in the manuscript on page 9, lines 180-182. In addition, Supplementary Figure S2 has now been updated to reflect the hypergeometric probabilities.

3. The correlations between DNA methylation and gene expression, as shown in Figure 3, are in general very weak, and whether the correlations tend to be positive or negative for different genomic elements is difficult to read from the figure since almost all medians are close to zero. It would be helpful to add the $y=0$ line to this figure and perform some statistical hypothesis testing (e.g. a Wilcoxon’s signed rank test) to evaluate the deviation of the medians from zero. It is also necessary to evaluate the significance of each correlation coefficient using more rigorous statistical tests instead of absolute values of correlation coefficients.

Thank you for your suggestions. Our correlation analysis was limited by the somewhat fragile buccal RNA, as well as the dynamic range of the expression array. We suspect that some of these

correlations would be stronger in other affected tissues that are not readily accessible for sampling (e.g. liver) or by using RNAseq (unavailable). The primary motivation for integrating the expression data was to make the case for a varied gene expression response (both increased- and decreased-expression profiles) in response to widespread hypomethylation at affected loci (page 7, lines 145-146).

We have added a dashed line at $y=0$ as well as two dotted lines indicating the thresholds beyond which the Spearman correlations between CpG methylation and gene expression were significant ($p=0.05$). The figure and figure legend have been updated accordingly (page 43, lines 960-961). A Wilcoxon signed rank test revealed that only those correlations involving CpG loci within 200 bp of transcription start sites (TSS200) and intergenic regions (IGR) differed significantly from zero ($p=0.027$ and $p=0.030$, respectively). The results of this Wilcoxon signed rank test have now been included in the manuscript on page 8, lines 157-159.

4. Figure 5 shows some genes associated with DMCs in ESAM. Since the correlations between DNA methylation and gene expression are very subtle, it is necessary to further show that the changes in DNA methylation are indeed coupled to changes in expression levels of these genes. Moreover, the genome browser tracks in this figure are not very informative. I recommend removing them or moving to supplementary figures.

As noted above, the primary intention behind the correlation analysis was to provide evidence supporting the notion that DNA hypomethylation is not merely a synonym for increased gene expression (as is often inferred). Buccal tissue is not known to be a particularly transcriptionally active tissue, and thus, whilst well-suited to our study of DNA methylation, is less well suited to a detailed study of the extent to which altered DNA methylation affects gene expression at all sites – this is a topic for future studies.

Of the four genes overlapping the regions shown in Figure 5, *MED24* was expressed at an appreciable level in buccal tissue, and this expression was strongly negatively correlated ($\rho = -0.645$) with methylation at the CpG locus cg23207054. This locus passed the Bonferroni threshold of significance in our single site methylation analysis, having a difference in mean beta-value methylation between ESAM and NESAM of -0.077 ($\sim 7.7\%$); we observed other similar examples in the dataset. We have now moved Figure 5 to the supplementary files as Figure S4 and have added Fig. 3C to show an example of the relationship between methylation and *MED24* gene expression values.

5. It is not clear how the list of 34 phenotypes associated with ‘kwashiorkor’ was generated based on HPO since searching for the term ‘kwashiorkor’ in HPO resulted in nothing. This needs more details to make sure that the readers can reproduce the results.

The 34 phenotypes associated with ‘kwashiorkor’ were retrieved from The Monarch Initiative (<http://monarchinitiative.org/>, accessed May 26, 2017) using links derived from *Groza et al.* (AJHG, 2015; ref. #34). These are now available as Supplementary Table S5 (page 9, line 188). In personal communication with Peter Robinson (senior author of *Groza et al.*), the phenotype ontologies for common diseases are undergoing additional curation and have been temporarily removed from the website; he did not indicate any systematic errors that would render the dataset incorrect, and, clinically, the terms are well matched to the Kwashiorkor phenotype.

6. Several terms need clear explanation. For instance, the slopes from meQTL analysis (Fig 6A)

are not explained in the manuscript. My guess is that they are the coefficients for the SNP term in the linear regression model but this needs clarification.

Thank you for pointing out this oversight. Yes, the slopes are indeed the β_1 coefficients for the SNP genotype term in the linear regression model. We have updated the figure legend to better reflect this explanation.

7. The authors have predicted the effects of nutrition-sensitive meQTLs on TF-binding and found that they are likely to alter the likelihood of TF-binding. This is interesting but a lot of necessary details are missing. It is unclear that what TFs are included in the prediction and whether other meQTLs (i.e. those nutrition-insensitive ones) are also able to change the TF binding affinities with a high chance.

Thank you for these shared (see reviewer #3) interests in the contingent TFs. In total, there were 117 unique TFs implicated by putative nutrition-sensitive meQTLs and 264 unique TFs associated with nutrition-insensitive meQTLs, of which 107 overlap. Indeed, the proportion of nutrition-insensitive meQTLs disrupting TF binding is similar to that of putative nutrition-sensitive meQTLs (0.65 and 0.64, respectively).

One hypothesis for nutrition-sensitive meQTLs involves differences in their transcription factor binding *affinity*. Although the proportions of single nucleotide variants (SNVs) that disrupt TF-binding were similar between nutrition-sensitive and -insensitive meQTLs, this could be the result of limited power to resolve all nutrition-sensitive meQTLs (i.e. some of the sites could turn out to be sensitive with larger sample sizes), and a limited knowledge of whether the extent to which SNVs affect TF binding affinity is comparable, particularly in the context of nutritional stress – i.e. an meQTL variant that reduces TF binding affinity by 50% under ‘normal’ conditions, could induce an even greater reduction in binding affinity under nutritional stress. Similarly, the underlying reference for TF binding affinity is not likely to have included states of nutritional stress, and thus several context-dependent sites may not be captured by the prediction algorithm (page 13, lines 272-275)

These results were included to raise the novel concept of nutrition-sensitive meQTLs (we are unaware of any previous descriptions thereof). As stated on page 16, lines 354-356, functional studies in cellular- and/or model-systems are needed to better understand these putative nutrition-sensitive sites. To aid such studies, we have added supplementary tables indicating which transcription factors’ binding sites are likely to be strongly influenced by putative nutrition-sensitive meQTLs (Table S8), alongside cataloging meQTLs without evidence of nutrition-sensitivity (Table S9). These tables are referenced in the manuscript on page 13, line 274 and page 31, line 690.

8. On page 13: ‘In fact, 40% of our differentially methylated sites showed strong correlation (Pearson $r \geq 0.7$) between buccal and blood DNA methylation (Methods)’. It is necessary to provide figures or datasets to support this statement.

Thank you for pointing this out. We have now added Supplementary Table S11, which we refer to in the manuscript on page 14, line 308, and have detailed this analysis in the methods section (page 25, lines 561-564).

Minor:

Font sizes in Fig 4 are too small to read.

Thank you for bringing this to our attention; the font sizes have now been increased.

Reviewer #2 (Remarks to the Author):

Schulze et al. analyzed genome wide methylation pattern in (a) severe acute childhood malnutrition (edematous SAM or ESAM) and in non-edematous SAM (NESAM). Transmethylation of 1-carbon substrates has been reported to be slower in acute, but not recovered, ESAM. So that DNA methylation changes could lead to the differences in acute pathogenesis between the two forms. Genome-wide DNA methylation was analyzed in 309 SAM children from two independent cohorts. ESAM, compared to NESAM, was characterized by significant hypomethylation at 99% of differentially methylated loci. Interestingly, adults recovered from SAM displayed no significant differences in methylation pattern between ESAM and NESAM. The authors claimed that gene expression and methylation at hypomethylated loci demonstrate both positive and negative correlations, so that a complex transcriptional response to the acute disturbance can be assumed. The authors showed that genes at hypomethylated loci also lit up in the Human Phenotype Ontology sub-phenotypes of kwashiorkor. They found enrichment for genes linked to nutrition and metabolism. Nearby genetic variation seems to influence methylation at these loci.

Major:

There are few previous studies pertaining to methylation pattern in severe malnutrition (e.g. references 68 and 69: C. J. Peter et al., DNA Methylation Signatures of Early Childhood Malnutrition Associated With Impairments in Attention and Cognition. *Biol Psychiatry* 80, 765-774 (2016); A. Sheppard et al., Molecular Evidence for Differential Long-term Outcomes of Early Life Severe Acute Malnutrition. *EBioMedicine* 18, 274-280 (2017)) These should be described in more detail in the introduction.

Thank you for your suggestion. The aforementioned studies compared DNA methylation between individuals with and without a history of SAM. In contrast to these previous analyses, we believe that a key feature of our study design is the comparison of DNA methylation between the edematous and non-edematous *forms* of SAM. These past studies have now been alluded to in the introduction (page 4, lines 80-83), but, for the sake of maintaining emphasis on the comparison between SAM types, we feel that the detailed contextualization of these studies should remain in the discussion (page 14, lines 295-297; page 15, lines 334-337).

Did the authors confirmation the methylation patterns with an independent method?

Yes, for three Jamaican samples analyzed in our study using the 450K array, we also assessed DNA methylation using capture-based bisulfite sequencing and found that there was a strong correlation ($r=0.95$) between methylation levels quantified by the two technologies (described on page 25, lines 561-564, and depicted in Supplementary Figure S8).

Minor

Abstract: it should be mentioned that buccal cells were used for the analyses.

Thank you; buccal cells are now mentioned in the abstract (page 2, lines 30-31).

Abstract: it should be mentioned that Illumina arrays were used.

Thank you; the use of methylation arrays is now mentioned in the abstract (page 2, lines 30-31).

Supplimentary Table S4: cg03637179 - p-value is not below 0.05, please remove color.

Thank you for bringing this point of confusion to our attention. The p-value for cg03637179 is indeed below 0.05, but only for beta-values and not M-values, which are listed in the last column. Since we only report on beta-values in the main manuscript, we have now removed the columns showing M-value results to eliminate similar future points of confusion. Additionally, we have corrected the spelling of “Supplimentary” to “Supplementary” for Tables S2 and S4.

Reviewer #3 (Remarks to the Author):

In “DNA Hypomethylation in Acute Kwashiorkor Provides a Molecular Link to Aberrant 1-Carbon Metabolism in Severe Malnutrition” by Schulze et. al., the authors compare the DNA methylation profiles of buccal epithelial cells from children with edematous and non-edematous SAM in two countries, as well as adults who recovered from SAM. The authors seek to link the clinical and metabolic differences observed between the two phenotypes with underlying epigenetic mechanisms. They demonstrate that almost all differentially methylated CpGs between the two cohorts are hypomethylated in ESAM, that nearby genes are associated with kwashiorkor-linked symptoms, and that genetic variation may influence the change in methylation. In contrast, there is no difference in DNA methylation between the subtypes in recovered adults.

Severe malnutrition is a critical issue worldwide, and this paper is both important and would be of interest to a broad readership. However, it is unclear whether the methylation effect size is biologically meaningful, and the last two results sections are difficult to parse without a schematic. However, although I have some critiques, I would recommend this paper for publication.

Major comments:

The mean absolute difference in methylation beta-values even for significant CpGs is very small (0.054). Is this biologically meaningful? The authors should consider introducing a threshold to identify sites with the strongest difference between groups.

The scope of our study, like that of most human association studies, does not readily allow for a definitive establishment of biological plausibility; however, there is some context that might be more suggestive of biological meaningfulness. First, we focused on significance thresholds that are more stringent than most other epigenomic studies (Bonferroni-adjusted P values and false-discovery rates < 1%; page 5, lines 100-104) and are thus likely to represent the largest differences between groups.

Second, the beta-value quoted above is a both mean value that includes larger effects. In addition, the same magnitude of DNA methylation change could have differing effects depending on which genes are affected in which tissues and the genomic context within which the CpG locus is found relative to the gene. For example, looking at two differentially methylated CpG loci, cg13272780 and cg23207054 with differences in mean methylation beta values between ESAM and NESAM of -0.046 and -0.077, the former, located within the 3' untranslated region of *RALBPI*, is moderately correlated with *RALBPI* expression ($\rho=0.395$,

$p=0.085$), while the latter, located within 200 bp of the *MED24* transcription start site, is strongly correlated with *MED24* expression ($\rho=-0.65$, $p=0.003$).

Third, it is worth considering that our study only provides a snapshot of hypomethylation at one time point after the initiation of disease – children who develop SAM are typically ill for days to weeks before receiving care, during which time they are subject to the cumulative effect of a prolonged, multi-locus insult – i.e. a shift in whole-systems biology that is likely to have some overall clinical/biological impact (page 16, lines 361-363).

It is unclear exactly how the authors performed the intra-individual comparison of CpG methylation to gene expression. They state that they compare each CpG to the expression of all genes within a 10kb region, but Fig 3B is divided by genic location. It is conceivable that a CpG in the promoter of one gene does not influence the expression of a nearby gene, which may explain why so many of the comparisons in Fig3B are not strong (rather than a “complex transcriptional response”). A table of the number and location of DMCs corresponding to each gene and vice versa would help resolve this.

Thank you for raising this important point. The reviewer is correct that we cannot tell, based on our data, whether the most proximal genes are also the most likely to be influenced by or associated with differential methylation. This is, however, a general problem of genome-wide associations (epigenetic or genetic). Standard practice in the field has focused on the most proximal genes, acknowledging the caveat that proximal genes may not be the ones most affected by differentially methylated loci. Closer examination of all the 151 probes used in our analysis of correlation between methylation and expression values, revealed that only five probes were closely associated with more than one gene highly expressed in buccal tissue. These five probes (cg01849531, cg02435083, cg2706273, cg08676173, cg16451995) are found in the same DMC (chr16:8,941,747-8,947,967), which overlaps the start of *PMM2* and the end of *CARHSP1*, which are the two genes with the expression of which each of the five probes was associated. Conversely, each of the 25 highly expressed genes from our correlation analysis was only associated with probes from a single DMC. With the exception of *PMM2/CARHSP1*, we are confident that in this analysis the methylation values are likely to be correlated with the expression of a single gene, and have added a corresponding statement to the revised manuscript (page 7, lines 152-154). We have also added a column to Supplementary Table S4 that lists the DMC for each CpG probe, so that the results described here can now be confirmed based on the data captured in Table S4.

Additionally, the authors should clarify whether CpGs over different genic features have different effect sizes.

Thank you for this suggestion. Based on a Kruskal-Wallis test ($p=0.047$), the effect sizes obtained from the single site linear regression analysis appear to indeed differ slightly by gene context annotation for CpG loci that passed the Bonferroni threshold of significance. More specifically, a Dunn’s test of multiple comparisons using rank sums revealed that the magnitudes of the effect sizes for CpG loci within 200bp of the transcription start site (TSS) were significantly smaller than those of CpG loci within 1,500bp of the TSS ($p=0.004$), gene bodies ($p=0.012$), 5’ untranslated regions (UTR; $p=0.019$), and the first exon ($p=0.022$). Conversely, the magnitudes of effect sizes of Bonferroni-significant loci found within 1,500bp of TSSs were larger than those of CpG loci in intergenic regions ($p=0.013$). We have updated the manuscript

with these results on page 7, lines 138-144, and have included an additional supplementary figure depicting these results (Fig. S1).

On Line 248, the authors state that the meQTLs are predicted to impact TF binding. Which TFs are implicated?

Thank you for sharing your interests in these TFs. As mentioned in our response to reviewer 1, in total there were 117 unique transcription factors implicated by putative nutrition-sensitive meQTLs. We have added supplementary tables to show which transcription factors' binding sites are likely to be strongly influenced by putative nutrition-sensitive meQTLs (Table S8) and meQTLs without evidence of nutrition-sensitivity (Table S9). These tables are referenced in the manuscript on page 13, line 274.

The authors do not speculate as to why slower carbon metabolism would specifically impact genes implicated in the presentation of ESAM. Are these genes poised in some way that makes them more sensitive to carbon metabolic changes than other sets of genes? The correlation between buccal cells and blood suggests that this is agnostic to tissue.

Our meQTL analysis was designed around the notion that genetic variation associated with differentially methylated clusters could potentially make some of the implicated genes more sensitive to changes in one-carbon metabolism/methyl-group availability during ESAM (page 16, lines 348-350). Definitive mechanistic insights into which particular characteristics of this set of genes makes them more 'poised' is an excellent question, but one that is beyond the scope of our current study.

More justification and analysis should be provided to address heterogeneous cell types in buccal samples.

When we first started our analysis the only available methods to quantify cell type heterogeneity were developed for blood samples; thus we used the tissue prediction feature of the Horvath methylation age calculator to assess the tissue origin of our samples. Since then, other methods, such as EpiDISH (PMID: 28193155), have been published. EpiDISH in combination with the EpiFibIC reference (PMID: 29693419) provides cell fraction estimates of epithelial, fibroblast, or immune cell origin from ENCODE designations (which do not include buccal epithelium specifically). Nonetheless, using the Robust Partial Correlations (RPC) method, we found "epithelial" as the major cell fraction in 91.6% (283/309) of acutely malnourished children, with no difference between ESAM and NESAM ($p=0.12$, Chi-square test). Conversely, "epithelial" was the major cell fraction in 27.7% (18/65) of adult samples, without differences in ESAM and NESAM ($p=0.82$, Chi-square test). These EpiDISH results are consistent with those from the Horvath tissue prediction, although EpiDISH forces a broader tissue-class prediction rather than a probability for specific cell types (as the Horvath algorithm does), which would be better for mixed tissue types (e.g. saliva + buccal). Thus our preference is for the Horvath designation, but we have indicated the results of the EpiDISH analysis on page 24, lines 540-544 and page 25, lines 555-557.

Minor comments:

- Figure 3A displays the results only for Malawian children. Why are the Jamaican children not included?

Thank you for pointing this out. The specification regarding Malawian children was misplaced. Figure 3A is in fact based on CpG probes located within differentially methylated clusters from the association study involving both Malawian and Jamaican children with SAM. Figure 3B, on the other hand, was only performed on Malawian children with SAM, since Jamaican RNA samples were not available. We have edited the figure legend accordingly.

- A schematic, including the number of ontology terms and genes included in each dataset, would assist in the interpretation of the section Differentially methylated loci converge upon genes related to nutrition and metabolism.

We agree, and have included Supplementary Figure S3 that clarifies the ontology analyses (page 9, line 193 and page 27, line 594).

- Line 157: Is the number of EFO terms informative? For instance, one gene with a very large number of linked EFOs, or several related EFOs, could skew this analysis.

As a way to protect against individual genes with very large numbers of associated EFOs driving the results in our analysis, we chose to report only those EFOs with two or more associated genes, and these are the ones displayed in Figure 4A.

- Line 251: What does this study investigate?

The aim of J. van Dongen et al.'s study was to investigate individual differences in DNA methylation in monozygotic twin pairs. We now mention this on page 13, lines 278-279.

- Line 506: Is there a percentage associated with “primarily”? Does it differ between subtypes? Were the samples that were predicted to originate from other sites excluded?

The term “primarily” refers to the predicted tissue with the largest probability from the Horvath predictor. For DC samples, all samples predicted to be buccal in origin had a median “buccal” probability of 0.84; the corresponding median “buccal” probability for DL samples was 0.3, suggesting a more mixed tissue (e.g. saliva + buccal). These values have now been included in the manuscript on page 24, line 538 and page 25, lines 552-553. There was no difference in “buccal” probabilities between ESAM and NESAM among DC samples ($p=0.7897$, Welch two sample t test) or among DL samples (0.7687, Welch two sample t test). Only a couple of samples did not confirm to these tissue predictions, and we did not exclude any samples on this basis.

Reviewer #4 (Remarks to the Author):

This is a thorough, fascinating study on the potential molecular mechanisms that may be implicated in the various physiological differences between the edematous and non-edematous forms of severe acute malnutrition. Given there is increasing awareness of the true prevalence of kwashiorkor (and that it is likely underestimated globally), combined with the fact that it remains unknown why kwashiorkor develops in some households / communities and not others, this is a relevant study of public health significance. Strengths of the study are many, for example, by obtaining DNA methylation samples from the patients during the acute phase of the illness, along with the integration of gene expression data.

At the outset I will be transparent that the focus of my review has been on aspects specifically

related to SAM, one-carbon metabolism and the public health implications. I defer to other referees with greater bioinformatics expertise for more insight on the methods.

Whilst I do think this is a novel study of interest to both epigeneticists and public health nutritionists, I have made some specific points below. In summary, my main concern is that the hypothesis and associated discussion sections appear to rest on a single study that does not support what you claim in the narrative. There is no evidence of decreased methionine transmethylation in children with edematous malnutrition in the referenced study. Therefore, I could suggest you place more prominence in your introduction and discussion on the literature showing reduced methionine concentrations in edematous malnutrition compared to marasmic patients. This could still feed into your rationale for investigating DNA methylation and would provide a stronger evidence base.

I found that in several places your language could benefit from minor editing to be more conservative in your statements, especially those involving causality. There were also some further instances where your references did not back up your arguments. Finally, I suggest that you edit your conclusion about future public health / clinical interventions for treatment of kwashiorkor, which I feel are currently premature at this stage.

I do think, however, that this study makes an important contribution to the understanding of how kwashiorkor is characterised at the molecular level, and trust that the below comments are relatively easy to address.

Specific points

Thank you, first and foremost, for your thoughtful comments and suggestions, we have addressed them in the sections below.

Introduction

Line 47: In many contexts where NESAM is treated, weight loss is often acute rather than chronic. ESAM is generally described as the presence of nutritional (or bilateral) edema to distinguish it from other (unilateral) forms. I suggest clarifying.

We have updated statement in the manuscript to now specify “bilateral, pitting edema” (page 3, lines 48-50).

Line 54: Consider removing ‘innate’, as this suggests the biochemical and physiological differences between the two forms of SAM are consistent between populations. Yet, as you have correctly mentioned in the same line, our understanding of the conditions are based on fragmented data that are far from cohesive.

“Innate” has now been removed.

Line 58: Again, I suggest you soften the language by removing ‘clear’. When we look at the hardest of clinical outcomes, mortality, it is actually not clear. In some countries kwashiorkor is associated with increased mortality compared to marasmus, but this is not a consistent pattern

and an area of continued research.
“Clear” has now been removed.

Line 59: I found the purpose of talking about the unchanged nature of SAM treatment for both conditions unclear here. Are you suggesting there should be different approaches? If so, the case should be made. In community management of acute malnutrition using ready-to-use therapeutic food there is very strong evidence that grades 1+ and 2+ of kwashiorkor can be treated successfully in outpatient settings, using the same protocol as for marasmus.

We thank the reviewer for this consideration. Although there have been significant strides in treating ‘milder’ kwashiorkor, there remains a line of argument that, as with sub-types of other diseases, there is some utility in tailoring treatments to differing pathophysiologies (e.g. Type I vs Type II diabetes). Children with kwashiorkor do indeed respond differently, and conceivably better, to current outpatient treatment than children with marasmus (Kabalo and Seifu, J. Health Popul. Nutr., 2017; PMID: 28279227), however, this was not always the case, and is not the case for in-patient or more severe grades of ESAM (e.g. Marasmic-Kwashiorkor; Trehan NEJM, 2013). That said, we also acknowledge the need to balance ‘lofty’ treatment goals with the real costs of their implementation. We have now included a statement to reflect this opinion, which also ties into the statement referenced in the section below (pages 3-4, lines 64-67).

Lines 60-62: As per comment above, it needs to be a bit clearer whether this section is general background or setting up your study rationale. It feels a bit out of place at present.

We have addressed this statement in combination with the above (pages 3-4, line 64-67). In pursuing this project we encountered criticism of funding attempts to molecularly understand SAM as a misuse of resources that could be used to ‘eradicate’ this long-standing problem. Our statement was included to indicate that whilst we do not wish to diminish public health efforts, we feel that it is important to understand the molecular basis of the different forms of SAM for a variety of reasons, including a general understanding of nutritional stress, as well as to inform potential treatment strategies; these reasons are now germane to nutritional disorders globally and should, ideally, be no different for those in lower middle income countries.

Line 63-64: Reference 24 is about protein turnover more generally and makes no claim to the specific components of 1-carbon metabolism. I suggest removing it.
We have removed the citation (page 4, line 70).

Line 64-65: You mention that in reference 26 there is a slower transmethylation (TM) of methionine in ESAM children during the steady-state re-feeding stage. Please check this. The authors state in the results, and again in the discussion, that ‘there were no significant differences in homocysteine remethylation, methionine transsulfuration, or methionine transmethylation between the groups at clinical phase 1’. Please also change the abstract (line 26) accordingly. This is a critical point for your paper as your rationale is centred on the hypothesis that a reduced transmethylation rate that could potentially lead to lower DNA methylation. The reference does not support this. Indeed, it actually suggests that despite lower protein breakdown the transmethylation rate is maintained.

We have now updated and elaborated on our initial statement (page 2, lines 26-27 & page 4, lines 71-73), to reflect that children with ESAM tend to have lower concentrations of methionine with slower rates of methionine total- (Table 1 of now ref 25) and methyl- (Table 2; ref 25) flux

compared to children with NESAM. Further, the derived (not directly observed) transmethylation of methionine is significantly reduced in the acute vs recovered phase for both SAM states (Table 2; ref 25), with ESAM trending towards a lower transmethylation in comparison to NESAM (though, as indicated by the reviewer, not statistically significant). The combination of these findings strongly suggests to us that the generation of methyl groups through one carbon metabolism is very likely to be aberrant.

Lines 70-71: Given the study (ref 26) did not show a significant difference in TM between the groups, I feel your causal statement about reversal of biochemical changes from dietary recovery is too strong.

Whilst Table 2 of current ref 25 did not show a significant difference in transmethylation between ESAM and NESAM, it did provide evidence for differences in transmethylation between the acute and recovered phase - there was a significant difference in methionine methyl-group flux that resolved upon recovery. Such a resolution of differences (a similar trend is noted with cysteine kinetics) between ESAM and NESAM is in accordance with the changes in DNA methylation that we see in recovered adults. We have, however, now tempered the statement in the manuscript (page 4, line 76).

Lines 71-72: “The phenotypic differences that characterize the two forms of SAM are only evident during acute malnutrition”. Wouldn’t your reference 69 (Sheppard et al. 2018) suggest otherwise?

Line 76-77: Given a number of 2014 studies (e.g. PMID 24980666, 24517147), alongside Sheppard et al. 2018 do suggest phenotypic differences in SAM-survivors by NESAM/ESAM, I was curious why you hypothesise that methylation differences would have dissipated by recovery? You mention one of these studies in line 316. It therefore seems you are basing your whole hypothesis on reference 26, which as mentioned above, does not seem to give sufficient evidence (of TM differences), even during the acute phase.

In response to the two preceding questions, the phenotypes seen in acute malnutrition do indeed resolve after recovery from the malnourished state; however, Sheppard et al. measured methylation in muscle (without cellular decomposition) from 41 recovered adults. Our working hypothesis is that in other tissues that are still developing in these growing children (with slower mitotic rates than the buccal tissue we interrogated), acute differences in DNA methylation might lag in their recovery and are thus more likely to be retained into adulthood; persistent differences in DNA methylation might then contribute to the complex disease phenotypes that others have observed in adult survivors of SAM (page 15, lines 339-344).

Results

I felt this section could benefit from being cut down if possible. Perhaps some of the details on the individual genes in lines 183-212 could be integrated in the discussion to ease the narrative flow?

A lot of methodological detail / justification was also included here, which could be edited down given you have a detailed methods section.

With respect to both of these critiques, given the breadth of analyses undertaken, we sought to provide sufficient methodological context and detailed results to meet the varied interests of a general audience of clinicians, public health gurus, geneticists, metabolism experts, etc, which is a delicate balance. This is reflected in the breadth of reviewers' comments, some of whom have asked for additional detail; therefore, we have thus chosen to maintain the current balance.

Discussion

Line 268: It is difficult for the reader to understand which types of studies you are referring to, so I suggest cutting the sentence.

We have added the citations to the studies we were referring to in this section (page 14, line 297).

Lines 272: This will need editing if you agree with my earlier comment that reference 26 does not report slower TM in ESAM.

The statement has now been updated (page 14, line 300).

Lines 279-280: Your reference 58 does not support your statement, and this will in any case very much depend on the genomic region. The Lowe et al. reference is comparing blood and buccal, and if anything emphasises tissue DMRs, not cross-tissue methylation similarities.

In response, we refer to the following quote from paragraph 2, pg 450 of Lowe et al.: *“Using unsupervised clustering we found two distinct clusters that separate all blood samples (including various different blood subtypes) and all other samples including buccals, stem cells, transformed cells, brain, kidney, liver and even sperm (Fig. 5)”*. The referenced Figure 5 shows a striking separation into two large clusters (generated based on 196,817 CpG methylation probes): one containing blood-derived samples and another with buccal samples alongside all others. We interpreted these findings as supportive of the notion that buccal samples are likely to be a more relevant proxy for less accessible tissues than blood would be.

Line 290: delete ‘acute’ as SAM is acute by definition. Same in line 29.

We have deleted “acute” in both instances.

Line 291: Soften the language here – to say a transcriptional profile is consistent with results from a referenced study suggests to the readers the studies are looking at the same topics and have similar results, but reference 59 is not about your topic specifically. Here you are simply stating that methylation-transcription dynamics are complicated, which you already mentioned in line 290 (so you could delete this sentence).

We have edited the statement to better separate our findings from those described in the referenced paper (page 14, line 319).

Line 302-305: I find this a little confusing since your hypothesis is on one-carbon metabolic components driving transmethylation rates, rather than energy sources (and dietary intakes for ESAM and NESAM are very often the same). Furthermore, the thrifty genotype hypothesis would presumably extend to ESAM as well as NESAM? It may be easiest to delete this section to avoid readers being distracted from your core messages, as you already have a lot of good content in the discussion.

Reference to the ‘thrifty gene’ hypothesis have now been removed. We do, however, see value in tying our molecular results to those observed with other commonly studied nutritional traits, especially given a broad audience. Therefore, we have clarified (rather than deleted) our statement regarding overlap with genes and pathways involved in obesity/diabetes (page 15, lines 331-333).

Line 338: This is very interesting. Please reference the studies you are referring to regarding different dietary intakes for those developing kwashiorkor.

This statement was initially included to highlight that kwashiorkor is rarely seen outside of low-quality plant-based diets. After consulting with our collaborators, however, we now feel this declaration, as stated, may have been misleading, considering that low-quality plant based diets are prevalent among both forms of SAM, with no significant difference between the two groups (Lin C.A. et al., *J Pediatr Gastroenterol Nutr.*, 2007). Thus, we have chosen to remove it from the manuscript (page 17, line 371).

Line 341: Throughout the discussion, and particularly here and lines 351-354, you have been strong in suggesting a putative causal association between aberrant 1-carbon metabolism, hypomethylation and the type of SAM. I suggest editing the language to be more cautious in reflecting the real possibility of reverse causation, which I do not feel you ever fully address. The intricate feedback mechanisms involved in 1-carbon metabolism suggest that changes in methionine supply are very well buffered, and may not actually affect DNA methylation (in addition to reference 26, see Nijhout et al. 2006, PMID 1799881, which would contest your hypothesis).

Thank you. We made every effort to avoid the terms ‘causal’ or ‘mechanistic’ in our work, and have updated the manuscript and title to (hopefully) remove any vestiges of implied definitive causation (page 16, lines 357-359). As noted, our study cannot definitively rule out the potential that the observed hypomethylation is a secondary or non-contributory finding; however, it is, in our opinion, difficult to divorce previous observations of methyl-flux and methyl-donors in SAM from our results (causal or not) – particularly as such widespread hypomethylation is rare amongst epigenomic studies. We now present this latter view as our ‘working hypothesis’. We also note that the buffering of methionine may be robust in cellular studies, but in the context of a widespread *in vivo* lack of macro- and micro- nutrients is it unknown whether such mechanisms will be nearly as robust or predictable.

Lines 345-6: Yes, the evidence base for lower methionine concentrations in ESAM is more widely replicated than your reference to altered TM rates in ref 26. I suggest you re-frame your rationale using these studies.

Per recommendation, we have rephrased our rationale (see previous responses to comments relating to current ref 25).

Line 356: Re-phrase, as choline and betaine are methyl donors. Relevant cofactors in 1-carbon metabolism, leading to effective methionine transmethylation and remethylation would include B6, B12, B2.

Considering the changes in DNA methylation that we see in our study, we believe treatment with methyl donors might be an effective way to support cells in mitosis, even though it may not address the specific, pathogenic defects in 1-carbon metabolism (page 17, line 381-387). This is

partially hinted at by recent mouse models of maize-vegetable diet- induced fatty liver (May, Nutrients, 2018)

Line 356-7: I do not think your suggestion of methionine supplementation is justified by your study at this stage. The focus of your study is on the potential molecular etiology of ESAM compared to NESAM. The reality in public health nutrition programming is that there is a very effective community treatment for kwashiorkor once detected. Are you suggesting that treatment is inadequate and that supplementation with methionine would work above and beyond current treatment protocols? If so, please justify. Furthermore, your reference 77 does not support your hypothesis – it was looking at a way to improve glutathione synthesis and did not test whether it was a viable way of preventing DNA hypomethylation. At most you can advocate for further research into epigenetic effects of methionine supplementation in patients with ESAM.

We agree with this statement. Our hope is that this study will contribute the foundational base needed to support research on the effects of methionine and methyl-donor supplementation in ESAM patients. We have changed the tone of the manuscript to advocate for further research.

Philip T James.

Reviewers' comments:

Reviewer #1 (Remarks to the Author):

The authors have addressed most of my concerns. However, there are still some remaining issues that have not been fully addressed. Specific points are listed below.

First, one evidence the authors use to support the association between DNA hypomethylation and ESAM pathology is the overlap between gene ontologies enriched in genes related to kwashiorkor and those enriched in the DMCs. While I agree with the authors that this result supports some extent of association between the DNA hypomethylation and kwashiorkor, I still think it is necessary and straightforward to directly evaluate the overlap between the two sets of genes instead of the enriched gene ontologies. If the overlap is significantly higher than random, that would be a stronger evidence to support the association between DNA hypomethylation and ESAM pathology. If the overlap is not significant then some discussion is necessary.

The authors have performed some additional analysis that shows that the fraction of meQTLs that affect TF binding is similar between nutrition-sensitive and -insensitive ones. Their interpretation of this result is that the failure to observe statistically significant difference between nutrition-sensitive and -insensitive meQTLs might be caused by undiscovered nutrition-sensitive meQTLs or other unknown mechanisms, which could be correct, but does not support their statement that nutrition-sensitive meQTLs are important because they can alter TF binding, because the nutrition-insensitive ones can as well. Since the authors have provided lists of TFs affected by the nutrition-sensitive and -insensitive meQTLs, some discussion about the difference in functions of the two lists of TFs would be helpful to clarify this point.

Minor:

Page 13, line 271: 'Fig 6B; 6C' should be 'Fig 5B; 5C'.

Reviewer #2 (Remarks to the Author):

The authors answered my remarks elegantly, no further comments. Anke Hinney

Reviewer #3 (Remarks to the Author):

I have assessed the revised manuscript and I am in general happy with the author's response.

Reviewer #4 (Remarks to the Author):

I thank the authors for their thoughtful and comprehensive response to my original comments. This remains a fascinating study. Where changes have been made to the manuscript in response to issues I raised, I am happy with the edits. The softened language around causality, the clarification of the findings as a working hypothesis, and the re-framing of the background literature on 1-carbon metabolism dynamics in children with oedematous malnutrition have all helped clarify the narrative hugely.

I also felt the authors brought out the the unique aspect of the research much more clearly in this revision. The findings of this study on the molecular aspects of oedematous SAM certainly do need to be published to generate discussion on what remains a complex and poorly-understood phenotype.

One minor point: line 26/27, I think you may have meant a 'reduced total methionine flux'.

Philip T James

RESPONSE TO REVIEWERS

Reviewer #1 (Remarks to the Author):

The authors have addressed most of my concerns. However, there are still some remaining issues that have not been fully addressed. Specific points are listed below.

First, one evidence the authors use to support the association between DNA hypomethylation and ESAM pathology is the overlap between gene ontologies enriched in genes related to kwashiorkor and those enriched in the DMCs. While I agree with the authors that this result supports some extent of association between the DNA hypomethylation and kwashiorkor, I still think it is necessary and straightforward to directly evaluate the overlap between the two sets of genes instead of the enriched gene ontologies. If the overlap is significantly higher than random, that would be a stronger evidence to support the association between DNA hypomethylation and ESAM pathology. If the overlap is not significant then some discussion is necessary.

In response to this comment, we randomly sampled a subset of genes equal to the number of significant genes. Each random sample was taken from amongst the >20,000 genes within 10kb of regions tested for differential methylation. For each of the 10,000 random samples, we determined the degree of overlap with the Kwashiorkor HPO genes. Our FDR significant genes had a substantially larger degree of overlap than random (z-score = 4.19). This information is included on page 9, lines 188-199 and shown in Supplementary Fig. S3. We are mindful, however, that this analysis was still limited by the small fraction of genes currently linked to HPO terms (fewer than 4,000 in the HPO build used in our study). Our ontology analysis thus provides additional valuable insight into which clinical ESAM phenotypes corresponded to hypomethylated DNA regions.

The authors have performed some additional analysis that shows that the fraction of meQTLs that affect TF binding is similar between nutrition-sensitive and -insensitive ones. Their interpretation of this result is that the failure to observe statistically significant difference between nutrition-sensitive and -insensitive meQTLs might be caused by undiscovered nutrition-sensitive meQTLs or other unknown mechanisms, which could be correct, but does not support their statement that nutrition-sensitive meQTLs are important because they can alter TF binding, because the nutrition-insensitive ones can as well. Since the authors have provided lists of TFs affected by the nutrition-sensitive and -insensitive meQTLs, some discussion about the difference in functions of the two lists of TFs would be helpful to clarify this point.

To clarify, alteration of TF binding was only considered as one possible/plausible mechanism through which nutrition-sensitive meQTLs might work. Our reference to 'undiscovered nutrition-sensitive meQTLs' really refers to potential misclassification between nutrition-sensitive and -insensitive sites, not to hidden/newly discovered sensitive sites. With larger sample sizes, and a more robust genome variation representation than on the array utilized, we anticipate that some of the current nutrition-insensitive meQTLs would shift into the category of nutrition-sensitive. In that case, if our consideration of TF binding was correct, we would anticipate that a large proportion of the TF binding insensitive meQTLs would be found to be misclassified. Testing this hypothesis, however, is beyond the scope of the manuscript. We have updated the paragraph (page 13, lines 286 to 292) to indicate those TFs that were unique to the nutrition-sensitive group and to clarify the implied potential for misclassification.

Minor:

Page 13, line 271: 'Fig 6B; 6C' should be 'Fig 5B; 5C'.

Thank you for pointing out this oversight; this has been corrected (now line 283 on page 13)

Reviewer #2 (Remarks to the Author):

The authors answered my remarks elegantly, no further comments. Anke Hinney

Thank you.

Reviewer #3 (Remarks to the Author):

I have assessed the revised manuscript and I am in general happy with the author's response.

Thank you.

Reviewer #4 (Remarks to the Author):

I thank the authors for their thoughtful and comprehensive response to my original comments. This remains a fascinating study. Where changes have been made to the manuscript in response to issues I raised, I am happy with the edits. The softened language around causality, the clarification of the findings as a working hypothesis, and the re-framing of the background literature on 1-carbon metabolism dynamics in children with oedematous malnutrition have all helped clarify the narrative hugely.

I also felt the authors brought out the the unique aspect of the research much more clearly in this revision. The findings of this study on the molecular aspects of oedematous SAM certainly do need to be published to generate discussion on what remains a complex and poorly-understood phenotype.

One minor point: line 26/27, I think you may have meant a 'reduced total methionine flux'.

Philip T James

Thank you for your generous and favourable response. We reviewed the data from the Jahoor paper and have corrected lines 26-27 to be more specific of methionine-related flux.

REVIEWERS' COMMENTS:

Reviewer #1 (Remarks to the Author):

I am satisfied.

RESPONSE TO REVIEWERS

REVIEWERS' COMMENTS: (response in blue)

Reviewer #1 (Remarks to the Author):

I am satisfied.

Thank you.